# Teaching hydrological modelling: Illustrating model structure uncertainty with a ready-to-use computational exercise

Wouter J. M. Knoben[1] and Diana Spieler[2]

[1]University of Saskatchewan Coldwater Laboratory, Canmore, Alberta, Canada
[2]Institute of Hydrology and Meteorology, Technische Universität Dresden, Dresden, Germany

**Correspondence:** diana.spieler@tu-dresden.de

**Abstract.** Estimating the impact of different sources of uncertainty along the modelling chain is an important skill graduates are expected to have. Broadly speaking, educators can cover uncertainty in hydrological modelling by differentiating uncertainty in data, model parameters and model structure. This provides students with insights on the impact of uncertainties on modelling results and thus on the usability of the acquired model simulations for decision making. A survey among teachers in the Earth and environmental sciences showed that model structural uncertainty is the least represented uncertainty group in teaching. This paper introduces a computational exercise that introduces students to the basics of model structure uncertainty through two ready-to-use modeling experiments. These experiments require either Matlab or Octave, and use the open-source Modular Assessment of Rainfall-Runoff Models Toolbox (MARRMoT) and the open-source Catchment Attributes and Meteorology for Large-sample Studies (CAMELS) dataset. The exercise is short and can easily be integrated into an existing hydrologic curriculum, with only a limited time investment needed to introduce the topic of model structure uncertainty and run the exercise. Two trial applications at the Technische Universität Dresden (Germany) showed that the exercise can be completed in two afternoons or four 90 minute sessions and that the provided setup effectively transfers the intended insights about model structure uncertainty.

## 1 Introduction

The ability to use computer models to provide hydrologic predictions is a critical skill for young hydrologists (Seibert et al., 2013; Wagener and McIntyre, 2007). Model use is so widespread that students will have to generate, use or present modelling results at some point in their professional career (Seibert et al., 2013). A very wide range of different models currently exists and it is arguably less important for students to learn how to use any specific model than to be taught general modelling concepts. Students should have some understanding of different modelling philosophies, learn to use different model types and be aware of the strengths and limitations of hydrologic modelling (Wagener et al., 2012). Given the societal need to provide

hydrologic predictions far into the future and the unknown (Kirchner, 2006), a core competence for young professionals is knowing how to provide such predictions in a scientifically sound manner.

Understanding of uncertainty in the modelling process is key to interpreting model results (e.g. Pechlivanidis et al., 2011; Blöschl and Montanari, 2010; Beven et al., 2011; Mendoza et al., 2015, among many others). Modelling uncertainties can be roughly classified as relating to the input and evaluation data, the estimation or calibration of model parameters, and the choice of equations that make up the model structure. These concepts should be an integral part of the hydrologic curriculum (Wagener et al., 2012; AghaKouchak et al., 2013; Thompson et al., 2012) in a teaching structure that includes student-driven, hands-on exercises that reinforce the taught concepts (Thompson et al., 2012). A survey among 101 teachers in the earth and environmental sciences (see Supplementary Material S.1) shows large differences in how much time is spent on teaching hydrologic modelling in general, whether model-related uncertainty is part of the course and, if so, which aspects of uncertainty are covered. Based on the survey, model structural uncertainty is the least represented uncertainty aspect in teaching. The main reason named for not covering model-related uncertainty is a lack of time, while the lack of good teaching materials is the second-most common explanation. Just 6% of respondents that did not cover uncertainty in their classes stated that the topic would be covered in another course.

Selecting a model that faithfully represents current and future hydrologic conditions in a given catchment is critical for realistic long-term projections of water availability. In other words, one requires "the right answers for the right reasons" (Kirchner, 2006). The difficult task of finding an appropriate model structure, i.e. the combination of which hydrologic processes are included in a model, which equations are used to describe these processes and how model states and fluxes are connected, can be referred to as model structure uncertainty and is a significant source of overall modeling uncertainty (Di Baldassarre and Montanari, 2009). Model structure uncertainty is being investigated with increasing numbers of models in increasingly varied selections of catchments (e.g. Perrin et al., 2001; Butts et al., 2004; Duan et al., 2006; van Esse et al., 2013; Knoben et al., 2020; Spieler et al., 2020) and results are consistent: model choice matters and selecting an inappropriate model for a given catchment can lead to simulations of questionable quality. Regrettably, suitability of a given model for the task at hand is not always the main driver in model selection. Prior experience with a given model combined with lacking insights into model strengths and weaknesses often lead to a certain attachment of hydrologists to their model of choice (Addor and Melsen, 2019). Hands-on experience with model structure uncertainty in a classroom setting, particularly through exercises that show that the choice of model can have a strong impact on the quality of simulations for a given catchment, will prepare students to think beyond their "model of choice". This will prepare students for when they will need to design modeling studies or interpret modeling results in their future careers.

Thoughtful interpretation of model results is among many other skills that are expected of young hydrologists (see for example Table 1 in Seibert et al., 2013). However, finding or creating course materials that cover all these expected skills and incorporating these materials into an existing curriculum is time-consuming, as is updating existing materials with new knowledge. This time is consequently not spent on preparing delivery of the material (Wagener et al., 2012). Wagener et al. (2012) therefore introduces the Modular Curriculum for Hydrologic Advancement (MOCHA), in which educators from many different countries freely share hydrologic course materials in a modular manner. Each module addresses a specific topic and

can theoretically be inserted into an existing curriculum with very little effort. Although the MOCHA project has been inactive for some time, the principle of freely shared, self-contained teaching modules can be of great use to the teaching community and is experiencing a revival in platforms such as HydroLearn (https://www.hydrolearn.org).

Seibert and Vis (2012) provide a stand-alone version of the *Hydrologiska Byråns Vattenavdelning* (HBV) model that is a good example of the MOCHA philosophy in practice. The software is specifically modified for teaching and comes with documentation and descriptions of various teaching goals. This is a so-called lumped conceptual hydrologic model that relies on empirical equations to describe catchment processes and on calibration to find its parameter values. Although there is debate about the usefulness of such models for predictions under change (see e.g. Archfield et al., 2015), there are good reasons to use

them as teaching tools provided that the limitations of these tools are clearly communicated to the students. Conceptual models tend to be much easier to set up and run than their more physics-based, spatially distributed counterparts; they generally have fewer lines of code and internal dynamics that are easier to grasp than those of physics-based models; and they continue to be widely used for practical applications. These characteristics mean that limited teaching time is spent on using and analyzing models rather than setting up the models; that students have more opportunity to explore internal model dynamics instead of

focusing on model outputs only; and that students obtain a firm understanding of the type of tools they are likely to encounter in positions outside of academic research (Seibert and Vis, 2012).

This paper introduces a set of computational exercises designed to give students hands-on experience with model structure uncertainty and to encourage critical thinking about how the results of a modelling study can be interpreted. Our goal is to increase the frequency with which model structure uncertainty is taught to (under-)graduates and to reduce the time investment

required for educators to do so. The exercises use two conceptual model structures applied to two carefully selected catchments to illustrate various important lessons about hydrologic model selection. Briefly, these lessons focus on the need to carefully interpret aggregated performance metrics, the dangers of applying models in new places based on performance elsewhere and the need to consider if a model's internal structure is an appropriate representation of the catchment at hand.

Wagener et al. (2012) outlines a need for multi-media tools that support teaching in hydrology, specifically mentioning

*"a model base with algorithms that the students can download and use to support their homework assignments or in terms projects (Wagener et al, 2004). Such algorithms need to be accompanied by sufficient documentation and data examples."* As such, this module uses open-source data to allow straightforward application in assignments and projects. Catchment data is obtained from the Catchment Attributes and Meteorology for Large-Sample studies (CAMELS, Addor et al., 2017), for the Middle Yegua Creek near Dime Box, Texas (ID: 8109700) and the Raging River near Fall City, Washington (ID: 12145500).

Models are selected from the Modular Assessment of Rainfall-Runoff Models Toolbox (MARRMoT, Knoben et al., 2019a) and are known as "Wetland model" (ID: m02) and "Collie River basin 2 model" (ID: m03).

The goals, learning objectives and materials of this module are described in more detail in Section 2. The exercises are described in Section 3 (ready-to-use student handouts can be found in Supplementary Materials S.2; source documents ready for adaptation by teachers can be found on the GitHub repository). Section 4 briefly summarizes the benefits of using these

exercises and describes two trial applications of this module at the Technische Universität Dresden, Germany. The exercises

require access to and knowledge of either Matlab or Octave. Course materials can be downloaded through GitHub: https://github.com/wknoben/Dresden-Structure-Uncertainty.

## 2   Description

This section describes the main teaching objectives (Section 2.1), the catchment data and models used (Sections 2.2.1 and 2.2.2 respectively), an overview of provided materials, requirements and install instructions (Section 2.3), and suggestions on how to integrate these exercises into an existing curriculum (Section 2.4).

### 2.1   Objectives and outline

The main goal of this teaching module is to facilitate teaching of model structure uncertainty in hydrology. Learning objectives are conveyed through comparative analysis between model results generated by students, using two conceptual model structures and two catchments and a single calibrated parameter set for each combination of model and catchment. The two models and catchments have been specifically selected from the results of a much larger calibraton exercise, in which 40+ models where calibrated to 500+ catchments (Knoben et al., 2020), for the lessons that can be conveyed by this specific combination of two models and two catchments. Note that detailed understanding of the selected models and catchments is not a goal in itself; they are only intended to convey the learning objectives specified in this section and were selected purely for that purpose.

A common way of evaluating a hydrologic model's performance involves calculating some aggregated score, such as the root-mean-squared error (RMSE), the Nash-Sutcliffe efficiency (NSE; Nash and Sutcliffe, 1970) or the Kling-Gupta efficiency (KGE; Gupta et al., 2009), that expresses the similarity between observations and simulations of a given state or flux (typically streamflow). In the remainder of this work we refer to the calculation of efficiency scores as the *accuracy* of a model's simulation, in the sense that simulations with higher efficiency scores more accurately resemble observations than the simulations from models with lower efficiency scores. This is contrasted by the term *adequacy*, which is more commonly used to refer to a model's degree of realism (see e.g. Gupta et al., 2012).

The models and catchments in this module are selected so that, after calibration of a single optimal parameter set per combination of model and catchment, in one of the catchments both models achieve very similar KGE scores despite the models having very different structures, while in the other catchment the models achieve very different KGE scores. This is intended to convey the following lessons to students (KGE scores and summary of these take home messages can be seen in Figure 1):

1. Model choice matters. Because all models are "hydrologic models" it is an easy assumption to make that the choice of model is largely one of taste or convenience, rather than one of suitability for the task at hand. Comparing the performance of both models in the Middle Yegua Creek catchment (ID: 8109700) shows that this is not the case: the choice of model strongly affects the accuracy of obtained simulations. In this particular case, the catchment experiences periods of no flow which model m03 can simulate but model m02 cannot.

2. Models with very different structures can achieve virtually identical efficiency scores in a given catchment. Comparing the performance of both models in the Raging River catchment (ID: 12145500) shows that both achieve similar KGE scores. Logically only one (or neither) of the models can be an appropriate representation of the hydrologic conditions in this catchment. This comparison shows that achieving relatively high efficiency scores in a given catchment is no guarantee that the model realistically represents the dominant processes in the catchment. In other words, high *accuracy* does not necessarily mean high *adequacy*.

3. Reinforcing the previous point, comparing the performance of model m03 across both catchments shows that the model achieves higher efficiency scores than model m02 in both places, while the catchments themselves are structurally very different (steep, humid and forested, versus flat, dry and bare; catchment descriptions are shown as part of the suggested exercises). Logically, model m03 may be a realistic representation of the hydrologic conditions in one of the catchments, but not both. This again shows that relatively high efficiency scores are no guarantee of the model producing "the right answers for the right reasons" (Kirchner, 2006).

4. Choosing a model based on past performance should be done with care. Comparing the performance of model m02 across both catchments shows that model performance is very different in both places and that having a "successful" model (in terms of having obtained accurate simulations) for one catchment is no guarantee that this model will produce accurate simulations in a different location.

Note that this teaching module does not cover the difficult issue of defining when a given efficiency score is high enough to consider the model as a plausible candidate for further consideration. This requires careful use of benchmarks (e.g. Garrick et al., 1978; Seibert, 2001; Schaefli and Gupta, 2007; Seibert et al., 2018) that dictate expectations for model performance, which is outside the scope of this module.

## 2.2 Catchments and models

### 2.2.1 CAMELS catchment data

The CAMELS dataset (Addor et al., 2017) provides meteorological forcing data and a variety of catchment attributes for 671 river basin in the contiguous United States. The catchments upstream of Middle Yegua Creek near Dime Box, Texas (USGS gauge ID: 08109700), and Raging River near Fall City, Washington (USGS gauge ID: 12145500), are used in this module. Our example hand-outs (Supplementary Materials S.2; GitHub repository) direct students to data that forms the basis of, and expands on, the catchment descriptions given below.

Middle Yegua Creek is a water-limited catchment (aridity fraction = 1.3, with the aridity fraction calculated as mean annual precipitation divided by mean annual potential evapotranspiration ) with a corresponding low runoff ratio (0.11; mean annual runoff divided by mean annual precipitation), low mean runoff (0.3 mm/d) and on average 30 days/year with no observed streamflow. Precipitation is sporadic (on average 294 days have < 1 mm precipitation) and mostly concentrated in autumn with

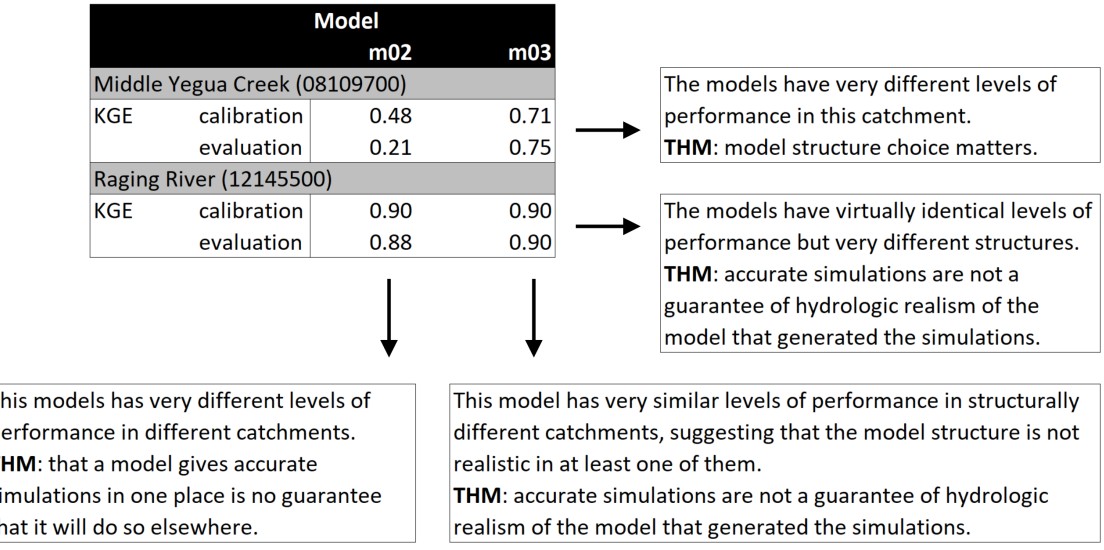

**Figure 1.** The table shows which calibration and evaluation Kling-Gupta Efficiency (KGE) scores the students will find when calibrating a single optimal parameter set for each combination of model and catchment, using their own adaptation of existing calibration code included in the MARRMoT repository. The calibration period is set as 1989-01-01 to 1998-12-31; evaluation data cover the period 1999-01-01 to 2009-12-31. Comparative assessments of these KGE scores transfers certain Take Home Messages (THM) about hydrologic model structure uncertainty, as detailed in the four text boxes.

little to no snowfall. The catchment is relatively large (615 $km^2$) with little variation in elevation (mean slope = 6 m/km). Vegetation cover consists mostly of cropland, shrubs and low trees.

Raging River is an energy-limited catchment (aridity fraction = 0.37) with a high runoff ratio (0.68), high mean runoff (3.9 mm/d) and observed streamflow on all days in the record. Precipitation occurs regularly (180 days with < 1 mm precipitation) and is winter-dominated although snowfall is rare (precipitation as snow fraction = 0.04). The catchment is comparatively small (80 $km^2$) and steep (mean slope = 86 m/km). Vegetation cover consists nearly exclusively of mixed forests.

The exercises use Daymet meteorological forcing data that is provided as part of the CAMELS data set (Newman et al.,
2015; Addor et al., 2017). Precipitation is part of the source data and time series of potential evapotranspiration are estimated using the Priestley-Taylor method (Priestley and Taylor, 1972). The exercises use data from 1989-01-01 to 1998-12-31 for model calibration, and data from 1999-01-01 to 2009-12-31 for evaluation.

### 2.2.2   MARRMoT models

The Modular Assessment of Rainfall-Runoff Models Toolbox (MARRMoT; Knoben et al., 2019a) contains Matlab/Octave
code for 46 conceptual models implemented in a single framework. Each model requires standardized inputs and provides standardized outputs. This means that data preparation and experiment analysis scripts have to be prepared only once, after which running and comparing different model structures becomes trivial. The toolbox is supported by extensive documentation,

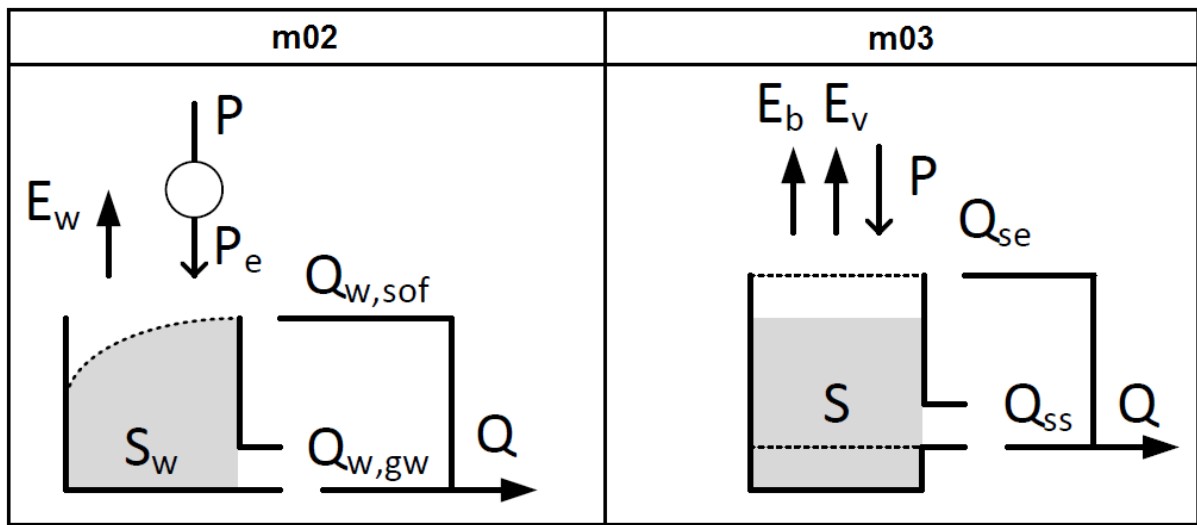

**Figure 2.** Wiring schematics of MARRMoT models m02 and m03 selected for this teaching module. Schematics are reproduced from Figures S2 and S3 in the Supplement of Knoben et al. (2019a) under CC BY 4.0. More in-depth model descriptions can also be found in this Supplement. Students are directed to these descriptions as part of Exercise 1.

divided into the main paper describing the toolbox setup; the Supplementary Materials to that paper describing each model, flux equation and default parameter ranges; a User Manual that provides guidance for practical issues such as installation, use and modification or creation of models and fluxes; and comments included as part of the actual computer code.

This course uses MARRMoT models m02 and m03 (names refer to consistent identifiers used in all MARRMoT documentation). Both have a single state variable and 4 calibration parameters but very different internal mechanics (Figure 2). Both models require time series of precipitation $P$ and potential evapotranspiration $E_P$ as input. Briefly, model m02 is part of the Flex-Topo approach (Savenije, 2010) and intended to represent the dominant hydrologic fluxes in a West-European wetland/riparian zone. This model has a single state representing catchment storage and uses 4 parameters to conceptualize interception by vegetation (turning precipitation $P$ into flux $P_e$), surface overland flow $Q_{w,sof}$ from a variable contributing source area, and groundwater flow $Q_{w,gw}$. Evaporation $E_w$ occurs at the potential rate. Model m03 is part of a study that develops a model for a semi-arid catchment in Western Australia (Jothityangkoon et al., 2001). This model also has a single state representing catchment storage but uses its 4 parameters to conceptualize a difference between bare soil evaporation $E_b$ and transpiration by vegetation $E_v$, saturation excess overland flow $Q_{se}$ if maximum storage is exceeded, and threshold-based subsurface flow $Q_{ss}$. Full details of both models, including state equations and flux parametrizations, can be found in the Supplementary Materials of Knoben et al. (2019a).

### 2.3 Materials, requirements and installation

#### 2.3.1 Provided course materials

All materials can be downloaded from https://github.com/wknoben/Dresden-Structure-Uncertainty. Provided are:

- Example exercise hand-outs, including instructions to obtain and install MARRMoT (hand-outs are also shown in Supplementary Materials S.2 for convenience);

- Prepared data for the second exercise (data for the first exercise are part of the MARRMoT install);

- An example script for model calibration for the second part of the exercise for use by educators;

- Calibrated parameter sets for both models that result in the KGE scores shown in Figure 1 for use by educators.

Note that these materials are sufficient to run the exercises with minimal effort. They do not include lecture materials to introduce the topic of model structure uncertainty. Such materials should provide a level of background and detail appropriate to the curriculum the exercises are inserted into, which will vary between curricula.

#### 2.3.2 Software requirements

Requirements for running MARRMoT are either Matlab with the optimization toolbox installed, or Octave. MARRMoT was developed on Matlab version 9.2.0.538062 (R2017a), with the Optimization Toolbox Version 7.6 (R2017a) and tested on Octave version 4.4.1 with the "optim" package (Knoben et al., 2019a). Note that the calibration workflow example (workflow example 4) differs slightly between Matlab and Octave 4.4.1 (see Section 7 in the MARRMoT User Guide on Github for more details about running MARRMoT in Octave). There are no differences in workflow example 4 between Matlab and Octave
5.2.0, thanks to a recent update to MARRMoT (M. K. Türkeri, personal communication, 2020).

#### 2.3.3 Install instructions

Detailed step-by-step install instructions for MARRMoT are included in our provided hand-out for Exercise 1. Briefly, download or fork and clone the MARRMoT source code on https://github.com/wknoben/MARRMoT. Next, remove the folder "Octave" if Matlab will be used. Open Matlab or Octave and ensure that all MARRMoT folders are added to the Matlab/Octave
path. MARRMoT is now ready to be used. Data necessary for Exercise 1 are part of the MARRMoT install. Data necessary for Exercise 2 are part of the GitHub repository that accompanies this paper and should be distributed to the students by the teacher.

### 2.4 Integration in existing curriculum

Assuming the existing curriculum provides access to and instruction in either Matlab or Octave, integrating these exercises
into the curriculum could happen along the following lines. The exercises would be preceded by a lecture that introduces the

concept of model structure uncertainty. We direct the reader to e.g. Perrin et al. (2001), Clark et al. (2011b) and Knoben et al. (2020) and the references therein for potentially useful sources to populate lecture materials with.

Next, our two proposed exercises can be run. Broad descriptions are provided in Section 3 while ready-to-use students handouts are included as part of the GitHub repository and in the Supplementary Materials (Section S.2). These exercises can be used as provided, or adapted to include more or different learning objectives. Distributing the data that underpins these exercises can either be done by referring the students to the GitHub repository that accompanies this manuscript, or by downloading the data and sharing these with the students in an alternative manner. Our example exercises include all instructions needed to obtain and install the MARRMoT source code. Students are then able to work through the exercises and will use MARRMoT to calibrate both models for both catchments, obtaining the Kling-Gupta Efficiency scores shown in Figure 1. Our proposed exercises contain guiding questions that will help the students draw the correct lessons from a four-way comparison of these scores, so that they arrive at the learning objectives outlined in Section 2.1.

Finally, a concluding lecture can focus on (1) how other sources of uncertainty can affect performance differences between models (see Section 4.1 for potentially useful materials), and (2) how to effectively deal with model structure uncertainty. Approaches to deal with model structure uncertainty could, for example, be (a) designing a model from the ground up for a specific combination of catchment and study purpose rather than relying on an off-the-shelf model structure (e.g. Atkinson et al., 2002; Farmer et al., 2003; Fenicia et al., 2016), (b) quantifying model structure uncertainty through the use of model inter-comparison (e.g. Perrin et al., 2001; van Esse et al., 2013; Spieler et al., 2020; Knoben et al., 2020), (c) setting more objective limits for when efficiency scores are considered acceptable by defining benchmarks that provide a context of minimum and maximum expected model performance (e.g. Schaefli and Gupta, 2007; Seibert, 2001; Seibert et al., 2018; Knoben et al., 2020), (d) attempting to define model *adequacy* through evaluation metrics that go beyond the use of aggregated efficiency scores that only measure *accuracy* and that rely on, for example, multiple metrics or data sources (e.g. Gupta et al., 2008; Kirchner, 2006; Clark et al., 2011a, b), or (e) applying model-selection or model-averaging techniques to effectively select or combine models with the appropriate strengths for a given study purpose (e.g. Neuman, 2003; Rojas et al., 2010; Schöniger et al., 2014; Höge et al., 2019).

## 3 Exercises

Sections 3.1 and 3.2 contain summaries of student handouts that can be used to run the two computational exercises. The student handouts themselves contain step-by-step instructions and guiding questions that take students from the comparison of KGE scores to the intended learning objectives. The full handouts can be found in section S.2 of the Supplementary Materials to this paper. To facilitate modification by educators, PDFs and LaTeX source files of the student handouts that describe these exercises are also available as part of the GitHub repository.

## 3.1 Exercise 1: MARRMoT basics

It is recommended to first run an exercise on an individual basis that introduces students to the MARRMoT framework. In the example exercise that is provided as part of the module's materials, students are asked to go through MARRMoT's four workflow examples and think critically about each example and possible ways to improve it. Download and installation of the toolbox are part of the exercise. The learning objectives for this exercise are for students to:

- Gain basic understanding of MARRMoT functionality;

- Be able to calibrate a hydrologic model and create diagnostic graphics that show the simulation results.

To achieve the learning objectives, students are asked to work through MARRMoT's four provided workflows. Workflow example 1 shows an example of running a MARRMoT model from scratch, using a single catchment and a single parameter set. The example includes loading and preparing climatic forcing, selecting one of the MARRMoT models to use, defining the models parameters and initial states, choosing settings for the numerical solver, running the model with the specified forcing and settings and analysis of the model simulations with the KGE objective function and qualitative plots. The predefined parameter values in this example are not well chosen and students are asked to vary the values and see how the simulations change. Manual sampling of parameter values naturally leads into workflow example 2.

Workflow example 2 replaces the arbitrarily chosen single parameter set with a random sampling procedure, using the provided parameter ranges that are part of MARRMoT. Results are visualized through qualitative plots. Students are asked to consider if a different model structure might not be more suitable than the pre-selected model and are directed to the MARRMoT documentation to investigate which other options are available in the toolbox. Students are asked to select a different model and re-run this workflow example, leading into workflow example 3.

Workflow example 3 shows how the code can be adapted to easily run multiple different MARRMoT models, with different numbers of parameters and state variables, from a single script. The example also includes code for visualization of the ensemble simulations. The example uses a randomly selected parameter set which is unlikely to give very good simulations. Students are asked to think of how to improve simulations and are asked to investigate the evaporation simulations as well as streamflow simulations.

Workflow example 4 shows an example of model calibration and evaluation and forms the basis for the model structure uncertainty exercises. Students are asked to adapt this script based on the code provided in workflow example 1 and 3 and are asked to consider better ways to initialize model storage values.

## 3.2 Exercise 2: Model structure uncertainty

This second exercise can be completed individually or in groups and gives the students hands-on experience with model structure uncertainty. The exercise asks students to calibrate two models (as introduced in Section 2.2.2) for two catchments (as introduced in Section 2.2.1), evaluate the resulting model simulations and think critically about the implications of their findings for model structure uncertainty. If working with groups, a possible approach would be to have each group work with

a certain combination of model and catchment first, and bring the groups together for a discussion of their findings after. Groups will reach different conclusions based on which model and catchment they were assigned and a class-wide discussion is critical to impart the take home messages of this module, because these can only be obtained by comparing the calibration and evaluation results across catchments and models (see Figure 1). The learning objectives for this exercise are for students to:

- Be able to navigate model documentation and the inner workings of hydrologic model code;

- Critically think about the relationship between model structure, catchment structure and model calibration and evaluation procedures, and in doing so arrive at the understanding outlined in Section 2.1.

As the first part of exercise 2, students are asked to familiarize themselves with the catchments and models. Catchment data are provided in the file "Part 2 - catchment data.mat" that is part of the course materials on GitHub. Students are asked to create some exploratory figures of the meteorological data and streamflow observations, and to take a look at the catchment descriptors that are provided as part of the CAMELS data set (Addor et al., 2017). Students are also asked to familiarize themselves with the two models by referencing the MARRMoT documentation and through an initial sensitivity analysis.

Next, students are asked to calibrate both models for both catchments, using Workflow example 4 as a basis for their code. This part of the exercise can take some time, partly due to the need to setup calibration and evaluation scripts and partly due to the time needed for the optimization algorithm to converge. This makes the second exercise well-suited for a homework assignment or for a brief introduction to the balance between accuracy of the optimization algorithm and its convergence speed.

Finally, guiding questions in the exercise description help students to compare their calibration results in four different ways:

1. By comparing the KGE scores of both models for the Middle Yegua Creek catchment (ID: 08109700), students are expected to find that an inappropriate model choice negatively impacts the accuracy of streamflow simulations. A key characteristic of this catchment is that flow events are separated by periods during which no flow occurs. Model m03 has the ability to simulate periods of zero flow due to it's threshold-based subsurface flow parametrization (see Figure 2), whereas m02 lacks this ability and will always generate some streamflow when modeled storage exceeds zero. As a result, model m03 achieves higher efficiency scores than model m02 does.

2. By comparing the KGE scores of both models for the Raging River catchment (ID: 12145500), students are expected to understand that models with very different internal mechanics can generate equally accurate streamflow simulations (in KGE terms). This is a well-known problem in hydrologic modeling (see e.g. Perrin et al., 2001; Knoben et al., 2020) and students should conclude that obtaining arbitrarily high KGE scores is no guarantee of hydrologic realism.

3. By comparing the performance of model m02 across both catchments, students will see that accurate model performance (in KGE terms) in one place is no guarantee that the model will generate accurate simulations in a different catchment. This comparison warns students of the risks associated with model selection based on legacy or convenience, rather than suitability for the task at hand (Addor and Melsen, 2019).

4. By comparing the performance of model m03 across both catchments, students are expected to realize that relatively high efficiency scores do not necessarily mean that the model structure realistically represents the dominant hydrologic processes in a catchment, because the model can logically do so for one catchment, but not both. This reinforces the lesson learned from the second comparison described above.

From this four-way comparison, students will gain the necessary insights to fullfil the learning objectives outlined in Section 2.1 and Figure 1. Students are asked to formalize these insights into three take home messages. A class-wide discussion can be used to reinforce this understanding.

## 4 Discussion

### 4.1 Notes on data, parameter and objective function uncertainties

This computational exercise lets students investigate the concept of model structure uncertainty in isolation, using a rather specific combination of data, models and parameter values. Our survey circulated among teachers in the hydrologic sciences (see Supplementary Materials S.1) indicates that model structure uncertainty is taught the least often in current curricula, whereas topics such as data uncertainty and parameter uncertainty are covered more regularly. Our suggested exercises implicitly assume that they are woven into an existing curriculum that introduces the students to other sources of modeling uncertainty and these are therefore not explicitly addressed in the provided hand-outs. To aid teachers in discussing the roles of other modeling-related uncertainties in the context of model structure uncertainty, we performed various brief analyses as outlined below.

### 4.1.1 Data uncertainty

Model calibration results are typically conditional on the data used and thus calibration outcomes can vary depending on how calibration and evaluation data are selected (see e.g. Coron et al., 2012; Fowler et al., 2016). This can be caused by differences in the statistical properties of calibration and evaluation data, leading to poor transferability of the calibrated model parameters to conditions different from those during the calibration period. More complex to quantify are measurement and estimation uncertainties that affect model input (e.g. precipitation, temperature, potential evapotranspiration estimates) and the observations the model is calibrated against (e.g. streamflow). To investigate the extent to which the findings in Figure 1 are conditional on our somewhat arbitrary choice of calibration and evaluation periods and hence on the idiosyncrasies of the data in either period and any statistical differences between both periods, we repeated the calibration exercise with reversed calibration and evaluation periods. Table 1 confirms that calibration and evaluation KGE scores are indeed conditional on the characteristics of the data used, but the relative performance differences remain consistent with those shown in Figure 1. There is a reasonable difference in calibrated model performance in the Middle Yegua Creek catchment and a substantial difference in evaluation performance in this place, whereas both models obtain very similar performance scores in the Raging River catchment.

**Table 1.** Kling-Gupta Efficiency (KGE) scores obtained using two different definitions of calibration and evaluation periods. The columns labeled "Period 1" refer to a calibration period that ranges from 1989-01-01 to 1998-12-31 and an evaluation period that covers 1999-01-01 to 2009-12-31 (i.e. values are replicated from Figure 1). The columns labeled "Period 2" show results obtained from calibrating the models on data from 1999-01-01 to 2009-12-31 and evaluating on data from 1989-01-01 to 1998-12-31.

| | | m02 | | m03 | |
| --- | --- | --- | --- | --- | --- |
| | | Period 1 | Period 2 | Period 1 | Period 2 |
| Middle Yegua Creek (08109700) | | | | | |
| KGE | calibration | 0.48 | 0.64 | 0.71 | 0.78 |
| | evaluation | 0.21 | 0.03 | 0.75 | 0.68 |
| Raging River (12145500) | | | | | |
| KGE | calibration | 0.90 | 0.93 | 0.90 | 0.93 |
| | evaluation | 0.88 | 0.89 | 0.90 | 0.88 |

### 4.1.2 Parameter uncertainty

Calibration algorithms are constrained in their ability to identify global optimal parameter sets by their termination criteria (e.g. maximum number of iterations, objective function convergence, parameter convergence, Pechlivanidis et al., 2011) and the complexity of the parameter/objective function response surface (e.g. Duan et al., 1992). As a consequence of a rapidly varying parameter response surface, small changes in model parameter values can lead to large changes in model simulations and thus to large changes in model performance. This is particularly prevalent when the model response surface contains discontinuities, as is the case for, for example, non-smoothed threshold-based flux equations (Kavetski and Kuczera, 2007). Collectively, these issues can be referred to as parameter uncertainty. We investigate the extent to which the KGE scores shown in Figure 1 are sensitive to these issues in two ways. First, we use a two-stage Latin Hypercube sampling approach (more details can be found in the caption of Figure 3; samples generated using the *SAFE* toolbox, Pianosi et al., 2015) to see if the calibration algorithm correctly identifies the general region where well-performing parameters may be found. Second, we select the 100 best performing parameter sets in each combination of model and catchment and compare the calibration and evaluation performance of those 100 sets to those of the calibrated values, to see whether small changes in the optimal calibrated parameter values would lead to large changes in KGE scores. Figure 3 (top four rows) shows that the calibration algorithm returns solutions in regions that match the regions of high KGE scores identified through parameter sampling. Figure 3 (bottom row) further shows that, if one were to use any of the top 100 parameter sets identified through Latin hypercube sampling, relative model performance is still consistent with that shown in Figure 1. There is a substantial difference in model performance in the Middle Yegua Creek catchment, whereas both models obtain very similar performance scores in the Raging River catchment.

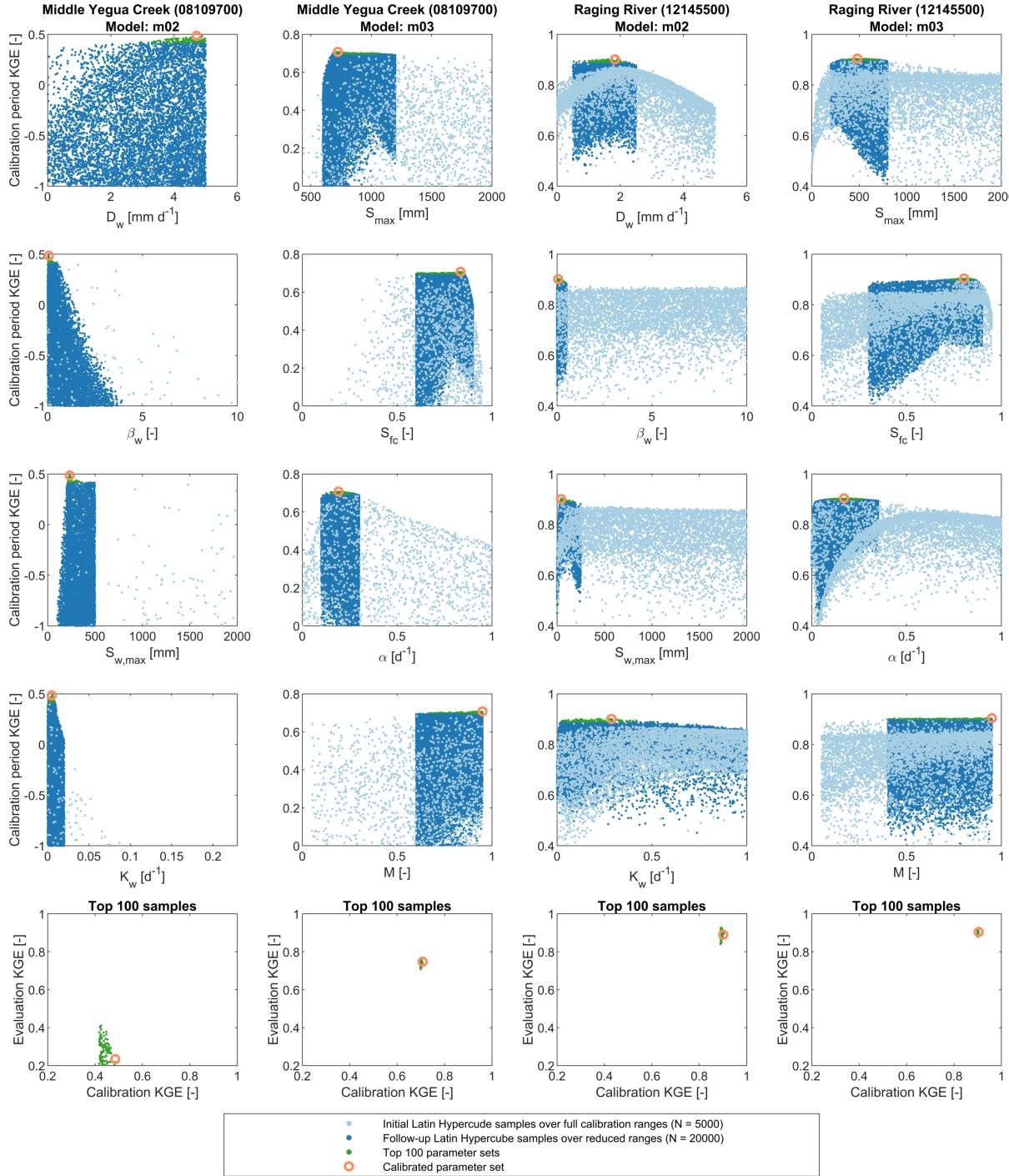

**Figure 3.** Top four rows: Latin Hypercube parameter samples for both models and catchments, and asscoiated KGE scores for the calibration period. Initial sampling (light blue) covered the full parameter ranges as used during model calibration to identify regions of interest for more thorough sampling (dark blue). See Figure S4 for a version of this figure with wider limits on the y-axes. Bottom row: comparison of calibration and evaluation KGE scores of the top 100 samples (defined as highest KGE scores for the calibration period) and calibrated parameter sets.

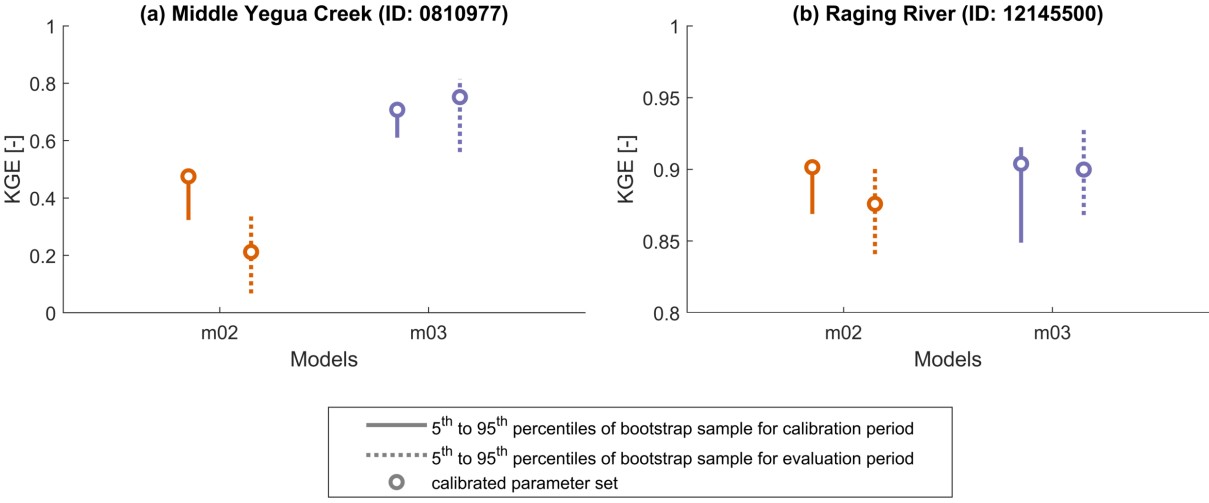

**Figure 4.** Sampling uncertainty in KGE calculation, quantified as the $5^{th}$ to $95^{th}$ percentile of the Bootstrap samples performed by the *gumboot* package. Note the different scales on the y-axes.

### 4.1.3 Sampling uncertainty in objective function calculation

The calculation of performance metrics such as KGE can be subject to considerable sampling uncertainty (Newman et al., 2015; Lamontagne et al., 2020; Clark et al., 2021), in the sense that differences between observed and simulated values on individual timesteps in a given time series can have disproportionate impacts on the overall KGE score. We quantified this uncertainty for the values presented in Figure 1, using the *gumboot* R package developed by Clark et al. (2021). Despite considerable uncertainty in the calculated KGE values, the main learning goals are unaffected: there is a distinct difference in model performance in the Middle Yegua Creek catchment (Figure 4a), while both models obtain similar KGE scores in the Raging River catchment (Figure 4b).

### 4.1.4 Choice of objective function

It is generally accepted that the choice of objective function has a strong impact on model calibration and thus on the simulations a calibrated model produces (see e.g. Mizukami et al., 2019, and references therein). While the Kling-Gupta Efficiency is derived from the components that make up the Nash-Sutcliffe Efficiency (Gupta et al., 2009), KGE and NSE scores are not directly comparable and their interpretation is quite different (Knoben et al., 2019b). We therefore repeated calibration of both models for both catchments with NSE as an objective function, to assess the extent to which our choice of objective function affects the four-way comparison outlined in Figure 1. Table 2 confirms that the choice of objective function has an impact on the actual performance scores obtained, but relative performance is consistent between KGE and NSE-based simulations: there is a distinct difference in model performance in the Middle Yegua Creek catchment with model m03 outperforming model m02, while both models obtain similar scores in the Raging River catchment.

**Table 2.** Calibration and evaluation Nash-Sutcliffe Efficiency (NSE) scores found when calibrating a single optimal parameter set for each combination of model and catchment, using data from 1989-01-01 to 1998-12-31 for calibration, and data from 1999-01-01 to 2009-12-31 for evaluation. The calibration and evaluation KGE scores listed in Figure 1 (obtained using identical periods for calibration and evaluation) are shown for comparison.

| | m02 | | m03 | |
| --- | --- | --- | --- | --- |
| | KGE | NSE | KGE | NSE |
| Middle Yegua Creek (08109700) | | | | |
| calibration | 0.48 | 0.29 | 0.71 | 0.44 |
| evaluation | 0.21 | 0.20 | 0.75 | 0.51 |
| Raging River (12145500) | | | | |
| calibration | 0.90 | 0.82 | 0.90 | 0.81 |
| evaluation | 0.88 | 0.85 | 0.90 | 0.86 |

## 4.2 Benefits

The main goal of the proposed exercises is to provide students with hands-on experience of the concept of model structure uncertainty. This section outlines various other benefits of using the course for both educators and students.

As mentioned, both models and catchments have been specifically selected out of much larger samples for the lessons they can convey. The exploratory work needed to do so (calibrating 40+ models for 500+ catchments) would typically be well outside of what is feasible for teaching preparation. With this selection already made, educators may spend their limited time on preparation of delivery of the course materials, without having to also spend time creating the exercises. The suggested exercises expose students to a variety of different concepts that easily transfer to other disciplines and topics, such as navigating peer-reviewed literature and model documentation, and working with open-source data, open-source software and version control through GitHub. Understanding of MARRMoT and model structure uncertainty can be leveraged into term projects or theses, providing students with a certain amount of modelling experience before their projects start. The recent publication of multiple CAMELS data sets covering the United States (Addor et al., 2017), Chile (Alvarez-Garreton et al., 2018), Brazil (Chagas et al., 2020), Great Britain (Coxon et al., 2020) and Australia (Fowler et al., 2021) can, for example, provide the necessary data for such projects.

## 4.3 Trial applications at Technische Universität Dresden (Germany)

The introduced computational exercises were run and evaluated in two different settings at the Technische Universität Dresden (TU Dresden). The first application was a workshop format in June 2019 where the attendees were both students (2 PhD & 4 MSc students) and academic or scientific staff (5). The intent of the workshop was to trial prototype exercises which could potentially be included in the curriculum of the first semester "Hydrological Modelling" module of the Hydrology

Master Program at TU Dresden. The content of the "Hydrological Modelling" module is an introduction to the possibilities and restrictions of representing hydrological processes with different model types; the creation, parameterization and application of abstract models; the objective assessment of uncertainties, and a critical examination of model results. Students are introduced

to basic MATLAB coding during the module and are expected to have seen and used a simple hydrologic model before.

An informal evaluation was conducted during this workshop by asking the attendees to fill in a short anonymous feedback form after the course was completed. Details on the workshop setup and the feedback procedure can be found in the Supplementary Material (S.3). Attendees unanimously reported that the course was easy to follow and complete, and that the main messages were clear. Various attendees specifically noted in their answers to open questions that the exercises were helpful for

better understanding the material covered during the preceding seminar on model structure uncertainty, confirming the importance of hands-on exercises to reinforce learning objectives (Thompson et al., 2012). The feedback of all attendees was used to refine and improve the exercise material.

The second application of the proposed exercises was delayed until January 2022 due to the COVID-19 pandemic. This time the exercises were offered in the intended first semester "Hydrological Modelling" module as an extracurricular exercise during

the last 4 weeks of the semester. Attendance to the four 90 minute sessions was thus entirely voluntary. Students were able to attend either in person or via a video conference.

The first session consisted of a lecture on model structure uncertainty, similar to the seminar given before the first workshop in 2019. During the second session, students downloaded and installed MARRMoT and individually started to work on Exercise 1 (See Section 3.1; consisting of MARRMoT basics on running a model, sampling parameters for a model and running several

models at once). During the third session students were introduced to model calibration with MARRMoT through the last assignment of Exercise 1 and started to explore and compare the catchment data for Exercise 2. They were asked to collect their insights by providing plots and notes on both catchments on a padlet (https://padlet.com/) to foster active collaboration between the in person and online attendants. During the last session students completed Exercise 2 (see Section 3.2) by adapting the previously used calibration script from Exercise 1 to the new catchment data. Each student was asked to choose at least

one model and one catchment and provide their calibration and evaluation results (KGE performance and modelled discharge) in a shared excel file. This allowed the lecturer to see when students were finished and to quickly plot some aggregated results for the following classroom discussion. During the final discussion students were asked to shortly present their results and compare them to the work of their peers. All insights were discussed and condensed to collectively formulate the final results at the end of the course.

This second workshop was accompanied by an updated feedback procedure consisting of two questionnaires. One questionnaire preceded the workshop and was intended to measure students' existing knowledge on uncertainty in hydrological modelling. The second questionnaire was handed out after the workshop and was intended to asses the changes in knowledge. Fifteen students initially showed interest in the extracurricular course and participated in the first survey while the final course assessment at the end of the semester was answered by 10 students. There were several questions that had to be answered on a

1 to 5 scale and some open questions in both surveys. All questions were asked anonymously via a google form and are listed in the Supplementary Material S.3

According to the results MARRMoT proved to be an easy-to-use tool (80% of participants found the course easy to follow/understand). All participants were able to download, install and use MARRMoT within minutes, using the instructions provided in the example handouts. MARRMoT's four workflow examples proved sufficiently documented for the students to quickly grasp the basic modelling chain (data preparation, model set up, model run, analysis of simulations) and satisfactorily complete Exercise 1. Only few Matlab related questions had to be answered by the course supervisor. Exercise 2 required students to set up and run their own model calibration scripts. Again all students were able to adjust MARRMoT's workflow examples with only minimal guidance and produce the expected results (note that a script showing a possible way to complete Exercise 2 is part of the provided materials for educators and available on GitHub). Figure 5 shows, that 90 % of students gained knowledge and confidence in the general area of hydrological modelling as well as in the area of model structural uncertainty. Not explicitly taught areas such as parameter and data uncertainty benefited as well from working through the general modelling workflow included in the four MARRMoT workflow examples. 80 % of students stated they have gained knowledge about the difference between hydrologic realism and high KGE scores and 90 % agreed to have gained knowledge on how certain model structures may be more suitable for certain places than others. Figure 6 shows that students improved their theoretical and practical knowledge of uncertainty aspects in hydrological modelling by comparing students knowledge of uncertainty before and after the MARRMoT course.

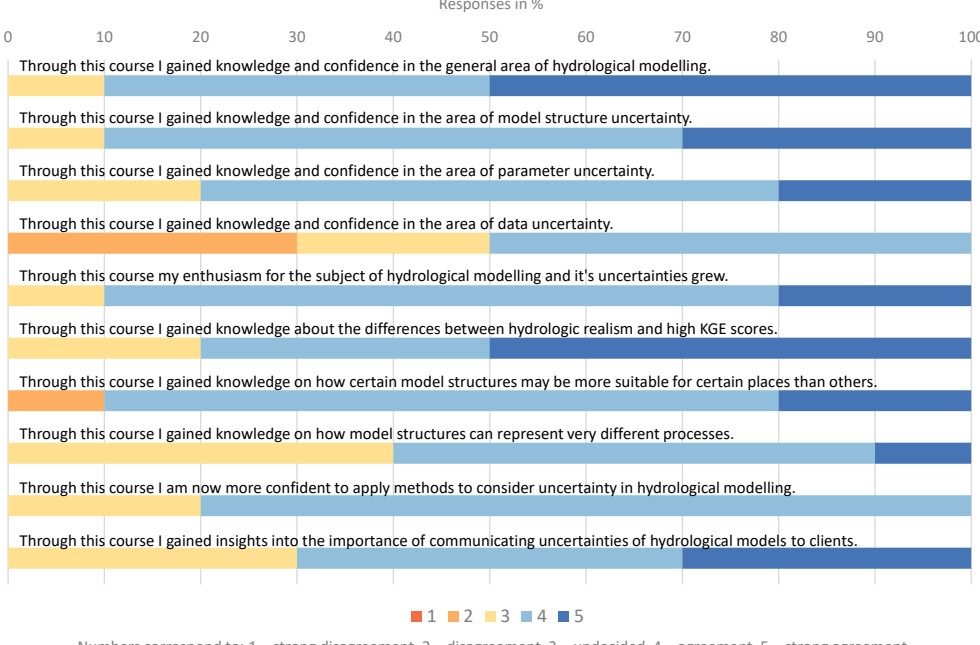

**Figure 5.** Responses to the feedback form distributed after the MARRMoT exercises. Only responses to the questions that had to be answered on a one to five scale are shown; responses to open questions are summarized in the Supplementary Material S.3. One indicates disagreement and five indicates agreement.

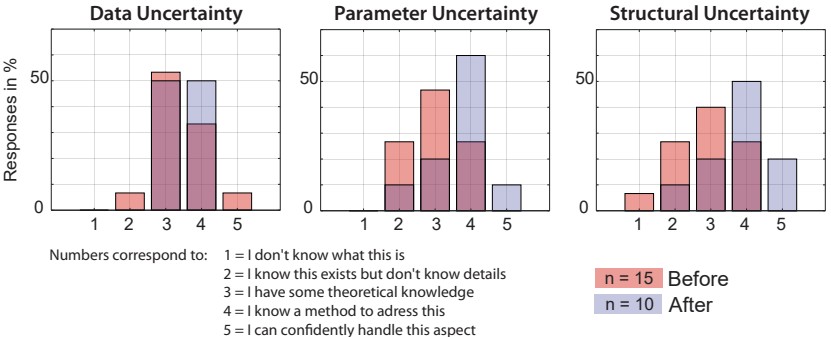

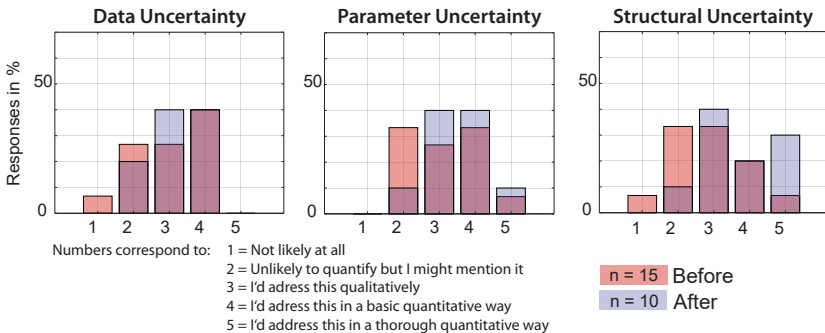

**Figure 6.** Comparison of responses to two questions evaluating student knowledge on different uncertainty aspects before and after the MARRMoT course. One indicates no knowledge and five indicates knowledge about theoretical and practical aspects of different uncertainty sources.

## 5 Conclusions

Understanding uncertainties in the modelling process is an important skill for graduates, and necessary to interpret the results from any modelling exercise. An informal survey circulated amongst educators in the Earth sciences suggests that model struc-

ture uncertainty is less often part of the curriculum than data and parameter uncertainty are. This paper introduces a set of ready-to-use computational exercises that can be used to introduce the concept of model structure uncertainty to students. Running the module requires either Matlab or Octave. The module uses open-source hydrometeorological data for two catchments and open-source model code for two models, specifically selected out of a much larger sample of catchments and models for the lessons these pairings can convey. Students are tasked to calibrate both models for both catchments and to evaluate the

calibrated models using data that was not used for calibration. Students are then asked to do a four-way comparison that will show that: (1) model choice matters, as in one of the catchments both models achieve very different levels of performance; (2) accurate model performance (in efficiency score terms) is no guarantee of hydrologic realism of the models, as in one of the catchments both models achieve very similar levels of performance, despite having very different internal mechanics; (3) the same applies when a single model produces accurate simulations (in efficiency score tmers) in two different catchments,

as logically the model may be realistically representing one of the catchments but not both; and (4) that the fact that a model produces accurate simulations in one catchment does not guarantee that this model will work well in another location, as the performance of one of the models is very different in both catchments. Trial applications of this module at the Technische Universität Dresden suggest that the module can effectively transfer these insights in the span of two afternoons or four 90 minute sessions. Data, model code, example exercise sheets and example code to complete the exercise are provided in a GitHub

repository so that educators wanting to teach model structure uncertainty can focus on the delivery of these materials, rather than on creating them.

*Code and data availability.*

Catchments used in this work are part of the CAMELS dataset (Newman et al., 2015; Addor et al., 2017) which can be downloaded from https://ral.ucar.edu/solutions/products/camels. Course materials can be downloaded from Github: https://

github.com/wknoben/Dresden-Structure-Uncertainty. The most recent version of MARRMoT can be downloaded from the "master" branch on Github: https://github.com/wknoben/MARRMoT

*Author contributions.* DS conceived the idea for the workshop that led to this publication, secured the funding and organized the workshop. WJMK selected the models and catchments. WJMK and DS created the course materials and wrote the paper. DS created the survey with help from WJMK. DS analyzed the survey results.

*Competing interests.* The authors declare they have no competing interests.

*Acknowledgements.* This work was partly funded by the European Social Fund (ESF) (grant 100270097). We are thankful for the Open Access Funding by the Publication Fund of the TU Dresden.

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
