# Peer review of "Teaching hydrological modelling: Illustrating model structure uncertainty with a ready-to-use computational exercise"

_Hydrology and Earth System Sciences, 2021_

## Author Response (AR1)

Response to the editor

Dear authors,

Two referees provided comments on your manuscript. Both appreciated your initiative of creating a tool that teachers could use to communicate hydrological model evaluation approaches to students. Both also raised concerns – or asked for clarifications – about how your study catchments and model variants were chosen (for inclusion in the manuscript), how model evaluation is performed and would be taught to students, the teaching materials (in general) that would accompany the module, and the easiness with which this module can be incorporated in "any" hydrology course.

I see, from your responses posted on the Interactive Discussion page, that you plan on - or have already started – thinking about how to address some of these comments. I therefore look forward to receiving and reading your revised manuscript (which will also be sent through another round of review). With best regards,

Genevieve Ali

Dear editor,

Thank you for your effort in managing this review process. We have updated the manuscript based on the recommendations of the reviewers and believe it has become clearer as a result. We speculate that certain comments by reviewer 1 about our choice to select two models and catchments ("why these?") can be traced back to a lack of clarity on our side about what we see as model structure uncertainty and why it matters. We have added a new paragraph to the introduction to set the scene for the remainder of the paper:

"Selecting a model that faithfully represents current and future hydrologic conditions in a given catchment is critical for accurate long-term projections of water availability. In other words, one requires "the right answers for the right reasons" (Kirchner, 2006). The difficult task of finding the right model structure, i.e. the combination of which hydrologic processes are included in a model, which equations are used to describe these processes and how model states and fluxes are connected, can be referred to as model structure uncertainty and is a significant source of overall modeling uncertainty (Di Baldassarre and Montanari, 2009). Model structure uncertainty is being investigated with increasing numbers of models in increasingly varied selections of catchments (e.g. Perrin et al., 2001; Butts et al., 2004; Duan et al., 2006; van Esse et al., 2013; Knoben et al., 2020; Spieler et al., 2020) and results are consistent: model choice matters and selecting an inappropriate model for a given catchment can lead to simulations of questionable quality. For a variety of reasons, the suitability of a given model for the task at hand is not always the main driver in model selection (Addor and Melsen, 2019) and it is not unlikely that students will encounter such cases in both academia and practice. Hands-on experience with model structure uncertainty in a classroom setting will prepare students for when they will need to interpret modeling results in their future careers."

We hope this provides the necessary context for the later part of the paper, where we select two models and two catchments to provide students with a hands-on example of various facets of model structure uncertainty. With an arbitrary selection of models and catchments, conveying these specific lessons cannot be guaranteed.

We have added a section on the evaluation of the exercises' effectiveness, using feedback forms obtained after our trial application. The intent of this trial was to see whether these exercises would form a useful addition to the Hydrology curriculum at TU Dresden. The trial application was therefore open to both students and faculty, to gather a wide range of opinions on how effective the exercises were. The trial had attendees ranging from MSc students to a full professor. Their comments (now included as part of the paper; see response to reviewer 1) suggest that our current selection of catchments and models was sufficient to convey the intended learning objectives and that the attendees saw no real need to include a larger sample of catchments or models.

Participation in the trial application was voluntary for students in their last year of the MSc level and everybody above. Eleven participants joined for the two afternoons, six of which were students and five of which were faculty members. This is a small number to draw conclusions from but compared to the typical number of people in the MSc Hydrology course at TU Dresden (usually between 8 and 15 students per cohort) this is a sizeable fraction of the maximum number of attendees our trial could possibly have had. Our intent was to re-run the course in 2020 and have it included as part of the Hydrology curriculum at TU Dresden in 2021, but the COVID-19 situation has caused substantial delays. We have added an evaluation of the module's effectiveness to the extent possible, based on the feedback forms from our trial application. We appreciate that this is not an ideal solution but see no easy way to gather responses from a larger number of students. The results from our trial suggest that the exercises effectively transfer the intended learning objectives. We therefore believe that making these exercises available to a wider audience, based on the current evaluation of their effectiveness, is likely more beneficial to educators than postponing this until some indeterminate point in the future at which we have a larger sample of evaluation responses.

Summary of other major changes to the manuscript:

- Changed the title to refer to a "computational exercise" rather than a "teaching module" to be more precise about what our contribution offers.
- Added a paragraph to the introduction to outline what we see as model structure uncertainty and why it matters.
- Rephrased "teaching module" in multiple locations to avoid implying that we also provide lecture materials. That educators will need to create lecture materials is now also explicitly mentioned in the "provided materials" section.
- Appended the "trial application" section with a description of the existing knowledge our students had and results from survey circulated among them to put numbers to the effectiveness of our exercises.
- Provided multiple clarifications as requested by the reviewers throughout the document.

Response to reviewer 1

**Major comments**

Selection of the two catchments in this study: I find the selection of the two catchments a bit problematic. Mainly, the two catchments vary in several aspects besides the so-called aridity fraction (e.g. size). This makes comparison difficult.

As a general note, both models and catchments were selected from a much larger sample that consists of 36 conceptual bucket models of varying degrees of complexity calibrated for streamflow simulation in 559 catchments (Knoben et al., 2020). Readers may of course use or include their own catchments (or decide to choose different models). The two models and two catchments used and presented here are however specifically selected to convey the lessons described in section 2.1 (Learning Objectives) through a four-way comparison. The fact that the catchments vary in multiple aspects is critical to two of the learning objectives we hope to convey, namely that (page 4, line 8):

*"Reinforcing the previous point, comparing the performance of model m03 across both catchments shows that the model achieves higher efficiency scores than model m02 in both places, while the catchments themselves are structurally very different (catchment descriptions are shown as part of the suggested exercises). This again shows that high efficiency scores are no guarantee of having used the "right" model."*

And (page 4, line 12):

*"Choosing a model based on past performance should be done with care. Comparing the performance of model m02 across both catchments shows that the model performance is very different in both places and that having a "successful" model for one catchment is no guarantee that this model will perform equally well somewhere else."*

We have added a note about the purpose of selecting these models and catchments, and a reference to where these learning objectives can be found to the start of section 2.1 (new text in bold):

"Both models and catchments have been specifically selected out of a sample of 40+ models and 500+catchments for the lessons that can be conveyed by each comparative exercise. **Note that detailed understanding of our selected models and catchments is not a goal in itself; they are only intended to convey the learning objectives specified in this section.** The catchments and models are described in sections 2.2.1 and 2.2.2 respectively."

We have also added an additional paragraph to the introduction that provides more background on model structure uncertainty, which hopefully provides a clearer context for our goal to convey general lessons, rather than specific insights about these models and catchments. See our reply to the editor.

Furthermore, one of the catchments reports zero-flows. Here it is important to note that the used model variants are by design not able to simulate zero-flows.

It is incorrect that our chosen models are not able to simulate zero flows. The threshold to flow generation in model m03 allows this model to produce zero flows (see also the figure below, obtained by running model m03 with data from catchment 08109700, showing zero flow simulations). That model m02 cannot produce zero flows is by design. Combined with using a catchment with occasional observed zero flows, this gives students a straightforward example of why choosing an appropriate model structure for a given catchment matters. This is key to the first learning objective (page 3, line 31):

*"Model choice matters. Because all models are "hydrologic models" it is an easy assumption to make that the choice of model is largely one of taste or convenience, rather than one of suitability for the task at hand. Comparing the performance of both models in catchment c08109700 shows that this is not the case: the choice of model strongly affects the accuracy of obtained simulations."*

Students are asked to follow this line of reasoning in the document that describes exercise 2:

*"Compare calibration and evaluation scores of both models for catchment 08109700 (Middle Yegua Creek). Based on what you know of this catchment's streamflow regime and the two model structures, which difference between the model structures do you think causes the difference in performance?"*

We will update the description of the first learning objective to include that the occurrence of zero flows and a model's (in)ability to produce these can be important. Changes in **bold**:

"Model choice matters. Because all models are "hydrologic models" it is an easy assumption to make that the choice of model is largely one of taste or convenience, rather than one of suitability for the task at hand. Comparing the performance of both models in catchment c8109700 shows that this is not the case: the choice of model strongly affects the accuracy of obtained simulations. **In this particular case, the catchment experiences periods of no flow which model m03 can simulate but model m02 cannot."**

[Figure]

The occurrence of zero-flows also makes the use of log-transformations for the computation of performance measures challenging.

Zero flows are indeed problematic for log-transformations. That said, our suggested exercises do not ask students to compute log flows and this is thus not an issue for the exercises as suggested. If one wishes to deviate from our suggested exercises and teach students how to log-transform flows, discussing the case of catchments with zero flows seems extremely relevant and having data from a catchment with zero flows readily available could be helpful with this. Therefore, we do not agree with the reviewer that the occurrence of zero-flows in one of these catchments is a major drawback to our setup and we do not agree with the implied suggestion that this catchment should be removed/replaced.

I am also a bit confused by the selection of the two model variants, why just these two?

As indicated in section 2.1, *"Both models and catchments have been specifically selected out of a sample of 40+ models and 500+ catchments for the lessons that can be conveyed by each comparative exercise."* These two models are sufficient to convey several important learning objectives to students. We will clarify at the start of section 2.1 that our purpose with this paper and these catchments & models is to convey general lessons about model structure and model evaluation, and that the two models and catchments should be seen as tools to achieve this. Detailed understanding of either models or catchments is not a goal in itself, beyond the understanding needed to grasp the learning objectives in section 2.1.

We have added a note about the purpose of selecting these models and catchments, and a reference to where these learning objectives can be found to the start of section 2.1 (new text in bold):

"Both models and catchments have been specifically selected out of a sample of 40+ models and 500+catchments for the lessons that can be conveyed by each comparative exercise. **Note that detailed understanding of our selected models and catchments is not a goal in itself; they are only intended to convey the learning objectives specified in this section.** The catchments and models are described in sections 2.2.1 and 2.2.2 respectively."

We have also added a brief summary of the learning objectives to the end of the introduction, so that we explicitly mention to the reader what our goal with this paper is before we introduce the catchments and models (new text in bold):

"The **exercises use** two conceptual model structures applied to two carefully selected catchments to illustrate various important lessons about hydrologic model selection. **Briefly, these lessons focus on the need to carefully interpret aggregated performance metrics, the dangers of applying models in new places based on performance elsewhere and the need to consider if a model's internal structure is an appropriate representation of the catchment at hand. The goals, learning objectives and materials of this module** are described in more detail in Section 2"

I am missing an evaluation of how successful the suggested module is. As it is now, basically the same claims that the authors make to motivate their module are also used to describe its success, which is not convincing. What would be needed is some form of evaluation by surveying students who took the class.

This is a very helpful suggestion. We circulated a survey among the students that included multiple questions to be answered on a 1-5 scale and three open questions. We currently summarize the response as a general statement that students found that the course *"was easy to follow and complete,*

*and that the main messages were clear. Various attendees specifically noted that the exercises were helpful for better understanding the material covered during the seminar, [...]"* (section 4.3, page 11, line 11). We will include an additional figure that shows responses to specific questions on the survey in an anonymized way and expand on the current discussion of the module's application at TU Dresden.

We now included a Figure to the Section "Trial application at TU Dresden" providing the responses to the feedback form distributed after the course. The figure shows the responses to the questions that had to be answered on a one to five scale. The three open questions asked about the main lessons that were learned in the course, what attendees liked or disliked about the course and some space for additional comments. The answers to the open questions are summarized as stated above. New text (changes in bold):

"The attendees were asked to fill in a short anonymous feedback form after the course was completed. **It had several questions that had to be answered on a 1 to 5 scale and three open questions. A summary of responses (4 MSc students, 1 PhD student, 3 Postdoctoral Fellows; senior faculty members provided verbal feedback) is shown in Figure 3.** Attendees unanimously reported that the course was easy to follow and complete, and that the main messages were clear. Various attendees specifically noted **in their open questions** that the exercises were helpful for better understanding the material covered during the seminar, showing the importance of hands-on exercises to reinforce learning objectives (Thompson et al., 2012)**. The number of models and catchments used in the exercises was sufficient and attendees were able to improve their understanding of the implications of model structure and parameter choice.** Various attendees also noted that the initial setup for sharing modelling results of Exercise 2 between the different groups was somewhat unwieldy. Consequently, the provided example handout for Exercise 2 is set up to work for an individual student and avoids the need to define groups and share results."

[Figure]

I am also missing information on how many students and with which background participated in the course in Dresden.

We will add a brief description of the hydrology curriculum at the TU Dresden and thus the expected background of the students who completed our survey to section 4.3.

We had 11 participants overall, of which 4 were MSc Students, 2 were PhD Students and 5 were faculty members. 8 attendees filled out the feedback form: 3 PostDocs, 1 PhD and 4 MSc Students. We now include this information about the participants in the manuscript text as follows (changes in bold):

"The course was attended by both students **(2 PhD & 4 MSc Students)** and faculty members **(5) outside of the regular curriculum. The intent of the course was to trial prototype exercises which could potentially be included in the curriculum of the "Hydrological Modelling" module of the Hydrology Master Program at the Technische Universität Dresden. The curriculum of the Hydrology Master at the Technische Universität Dresden covers the application of numerical tools for the planning and management of hydrological and water management systems, planning and implementing measurement networks and campaigns, data analysis, working with geographic information systems and various modelling techniques. At the point when this course was/would be held, students are expected to have some basic coding experience and have seen and used a simple hydrologic model before."**

The authors claim that their module could be added into 'any hydrology course with minimal effort' (P10L10). I'm afraid I have to disagree for several reasons:

1. If at all, then it can be added to courses in hydrological modelling, but not all hydrology courses.

Regarding point 1, we make the implicit assumption that modelling is part of most hydrology courses. We realize that this is not necessarily correct and will rephrase the manuscript accordingly.

This comment has resulted in numerous small changes throughout the manuscript.

2. If Matlab is not used in a particular class, including this module is by no means trivial

Regarding point 2, the prerequisite that "the module requires either Matlab or Octave, […]" already appears:

- In the abstract (page 1, line 10)
- At the end of the introduction (page 3, line 3)
- In the description of the MARRMoT framework (page 5, line 24)
- In the section "Software requirements" (page 7, line 10)

We will add this statement to the conclusion section as well to ensure this message is present in all locations a reader is likely to look for it. We will rephrase any sentences that talk about inclusion into existing curriculums as well to mention the need to have access to Matlab/Octave. This should provide the reader with enough information to judge whether they want to implement this module. To emphasize the need for a certain background in computing skills, we have:

- Reorganized the abstract slightly so that the need for Matlab or Octave is more prominent;
- Added the need for Matlab/Octave to the conclusions.

3. Teaching materials are not provided; this would be important as a service to a potential teacher who wants to adapt this module in their course.

Regarding point 3, teaching materials are provided on GitHub, as indicated in the introduction, section 2.3.1 and the section *Code and data availability.* We will clarify that these materials do not include lecture slides, but they do provide:

- Prepared data (meteorological time series and data describing both catchments);
- Pdf's and LaTeX source files for the two exercises described in section 3;
- An example script showing how to complete exercise 2;
- Calibrated parameter sets for all combinations of models and catchments, resulting in the calibration and evaluation results shown in Table 1.

This is sufficient to run the suggested exercises with minimal effort and for adaptation with new models or data by a teacher wishing to do so. If the reviewer disagrees, we would welcome more detailed comments about what they think is missing. We amended the text to acknowledge that an educator will need to spend some time on creating introductory lecture materials:

- In the abstract (new text in bold):
  "The exercise is short and can easily be integrated into an existing hydrologic curriculum, **with only a limited time investment needed to introduce the topic of model structure uncertainty and** run the exercise";
- In the section "Provided course materials" (new text in bold):
  "**Note that these materials are sufficient to run the exercises with minimal effort. They do not include lecture materials to introduce the topic of model structure uncertainty, because such materials should logically connect to the curriculum the exercises are inserted into.**
- In the new section "2.4. Integration in existing curriculum"

4. The fixed selection of catchments and models might limit the utility of the module.

Regarding point 4, summarizing our earlier responses, these models and catchments were selected for the general lessons they can convey. We believe this is a good introduction into several important aspects related to model structure uncertainty. Given that both CAMELS data (for multiple countries) and the MARRMoT toolbox are freely available, those wishing to give their students an expanded experience can easily do so. We hope our revisions to the text (discussed in responses to other comments) have sufficiently clarified that our goal is not to teach specific understandings about catchments A and B and models X and Y, but to transfer general lessons about model structure uncertainty.

I would recommend describing the module first in generic terms. Both catchments and model variants could be left open to be selected as appropriate for a particular course. Forst of all, there is great value in using catchments that the students are familiar with. Using US catchments might not be the most pedagogical choice in many cases.

Briefly, this is true if one is trying to teach locally relevant hydrologic understanding. As argued before, for general understanding of difficulties relating to modelling, specifically selecting a few catchments and models precisely for their ability to convey this general understanding seems logical to us. Our trial application for a German audience suggests that using catchments in the United States was not detrimental to conveying these learning objectives and appetite among the audience for inclusion of more catchments was low.

While finding local examples that show our intended learning objectives would be great, this takes effort on part of the teacher (not to mention that local data or models may not be available at all). Our goal is to reduce the initial effort needed to teach these concepts at all. Those who wish to go beyond our provided setup are of course welcome to do so.

Furthermore, depending on which programming language/modelling frameworks are used in a course, it might also be more useful to use an alternative to the option presented here.

In cases where Matlab or Octave are not available, a teacher is of course welcome to simply use the learning objectives presented in our paper if they think them relevant enough to include. For those that do teach Matlab (as is still common in universities), this paper presents a useful tool. Teachers/Students requiring an open source software may also use the Octave code provided in the MARRMoT framework, avoiding the need for (expensive) licenses.

In a second step, a concrete implementation of the module could be described (=as it is described now) and guidance could be given on alternatives.

We assume guidance on alternatives is meant in the context of the comments above, e.g. meaning how to use different catchments, models or programming languages. This seems so broad to us that any guidance will be either obvious (e.g. "one can look for data from local catchments instead") or unhelpful (e.g. "if Matlab or Octave are not available, one can consider converting this exercise to their programming language of choice").

We appreciate the reviewer's insights on how to make this course more generally applicable, but we expect that readers will be able to make the adaptations the reviewer outlines without having us guide them through the process. We prefer to have the paper focus on what we do have (a set of helpful exercises for a given programming language), rather than attempt to outline how educators could run their own courses in a more general sense.

Finally, it is crucial to evaluate the module in some way (e.g. student survey before-after)

Agreed, and we will add this to the extent possible.

As mentioned in a response earlier we circulated a survey among the attendees after the course. The responses are now included as Figure 3 in the manuscript. The results show that the course was easy to follow, and the messages were clear. The answers to the question "Were you aware of the implications the choice of model structure may have before the course" suggest that half of the participants improved their awareness of the importance of model choice. Given that part of the audience were faculty members who may be expected to already possess some understanding of model uncertainty, this seems to indicate a successful knowledge transfer.

**Minor comments**

P3L26 (mathematically) accurate – I think you just mean 'better'. Note that a model can be mathematically accurate but still totally useless.

The choice of "(mathematically) accurate" was deliberate, to convey that a high NSE or KGE score does not necessarily mean a hydrologically useful model. To us, "better" implies the latter more than it does the former. This is already made explicit in the text about learning objective 2 (copied here for convenience). To avoid confusion we have removed the word "mathematically".

"Models with very different structures can achieve virtually identical efficiency scores in a given catchment. Comparing the performance of both models in catchment c12145500 shows that both achieve similar KGE scores. Logically only one (or neither) of the models can be an appropriate representation of the hydrologic conditions in this catchment. This comparison shows that achieving high efficiency scores in a given catchment is no guarantee that the model accurately represents the dominant processes in the catchment."

P5L10 Aridity fraction: please explain this term and how it is computed

We will add this as requested. New text (changes in bold):

"(aridity fraction = 1.3**, with the aridity fraction calculated as mean annual precipitation divided by mean annual potential evapotranspiration**)"

P8L2: The statement that instructions are straightforward is followed by a 'fork and clone' statement that might be not at all straightforward to most readers.

We will clarify that more detailed instructions regarding the forking and cloning of Github repositories are provided as part of our suggested exercise 1. New text (changes in bold):

"**Detailed step-by-step install instructions for MARRMoT are included in our provided text for Exercise 1. Briefly**, download or fork and clone the MARRMoT source code on https://github.com/wknoben/MARRMoT**"**

P10L4: formalize? Do you mean formulate?

We mean this in the sense of "formalize your thoughts" by writing them down.

P10L30: Does this mean it was an one-day course in practice?

Yes. We will rephrase this. New text (changes in bold):

"This course was run **over the span of two afternoons** at the Technische Universit\"at Dresden (Germany) during June 2019."

P11L16: sorry, but the choice of one single student can't be really used as a convincing argument

Interestingly, the number of students is now two. That said, we agree that this is not necessarily an outcome of the teaching module presented in this paper and we have therefore removed this sentence.

Figure 2 is hard to read and needs to be improved. I am olso a bit wondering about the shown precip data, for me it does not look as if "on average 294 days have < 1 mm precipitation" from this figure

This is an unfortunate consequence of squashing 20 years of data into a few centimeters of graphic. We will consider the usefulness of this figure and change or remove as appropriate.

This Figure has now been removed from the manuscript.

Response to reviewer 2

This manuscript explains a module prepared by the authors to teach students the concept of the model structural uncertainty. Along with description of the module, results of the survey designed by the author are discussed to show that the amount of time and effort put toward teaching this concept has been minimal among teachers in the earth and environmental sciences. I found the manuscript very well-written and easy to understand. Moreover, making the teaching module ready for other teachers is a big plus, making the work a potentially popular study among the community of hydrology teachers.

Thank you for these kind words. It is good to know you see merit in our work.

However, the manuscript does not appear to fall under the scope of HESS that looks for studies that "contribute to the advancement of hydrological modelling, hydrological monitoring and data analysis, process concepts, experimental design and technology, or theoretical foundations".

We agree that our manuscript does not fit well into the main research category HESS looks for. However, HESS does accept manuscripts related to education and outreach (https://www.hydrology-and-earth-system-sciences.net/about/manuscript_types.html) and we have submitted our paper in this category.

Moreover, I have the following two major comments can help authors improve their manuscript and making it more easily adaptable for other teachers.

Both are discussed below.

Students require clear directions on how to evaluate the models, but the manuscript does not discuss which directions should be given to students to evaluate the uncertainty. It is mentioned on page 8 line 25 that qualitative plots are used to visualize the results, but it is not clear what those plots are. Also, KGE was used as the calibration objective, but calibration is inherently a multi-objective optimization task. Therefore, I encourage the authors to discuss with more in-depth information about what directions should be given to students in this course to be able to evaluate the uncertainty.

This is implicitly discussed in the exercises we propose (these can be found on the GitHub page) and the workflow scripts that accompany the MARRMoT toolbox, but we agree that this can be clearer in the main manuscript. We have added the clarification that this explanation is part of our proposed exercise sheets that can be found on Github to the start of Section 3, and have added the same information to the Supporting Information of this paper for convenience.

I believe the big missing pieces of puzzle in the module are:

- What should students do when they learn the fact that the model structural uncertainty exists? For example, should they discard all models but one? Or, should they select a sub-set of the models?
- How could students incorporate the estimated model structural uncertainty in their studies? For example, could they come up with a probabilistic estimation of the system response to hydrologic events?

Both are good points. We will add a new section "2.4 Proposed integration in existing curriculum" to discuss these points. This section would cover:

- Work needed by a teacher to integrate this module into their classroom beyond the materials we provide; i.e. introduce the topic of model structure uncertainty to students before they start the exercise (this addresses a specific comment by reviewer #1).
- Provide a brief overview of how model structure uncertainty can be quantified during the exercises and connect this to our proposed exercises (the 1st point the reviewer makes).
- Provide a brief overview of methods that have been used to deal with this resulting uncertainty (the 2nd point the reviewer makes), which can be discussed after the exercises.

New section:

"2.4    Integration in existing curriculum

Assuming the existing curriculum provides access to and instruction in either Matlab or Octave, integrating these exercises into the curriculum could happen along the following lines. The exercises would be preceded by a lecture that introduces the concept of model structure uncertainty. We direct the reader to e.g. Perrin et al. (2001), Clark et al. (2011b) and Knoben et al.(2020)  for potentially useful sources to populate lecture materials with.

Next, our two proposed exercises can be run. Broad descriptions are provided in Section 3 while ready-to-use students handouts are included as part of the repository and in the Supplementary Materials (Section S5). These exercises can be used as provided, or adapted to include more or different learning objectives. Distributing the data that underpins these exercises can either be done by referring the students to the GitHub repository that accompanies this manuscript, or by downloading the data and sharing these with the students in an alternative manner. Our example exercises include all instructions needed to obtain and install the MARRMoT source code.  Students  are  then  able  to  work  through the  exercises  and  will  use  MARRMoT  to  calibrate  both  models  for  both catchments, obtaining the Kling-Gupta Efficiency scores shown in Figure 1. Our proposed exercises (see Supplementary Materials S5) contain guiding questions that will help the students draw the correct lessons from a four-way comparison of these scores, so that they arrive at the learning objectives outlined in Section 2.1.

Finally, a concluding lecture can focus on how to effectively deal with model structure uncertainty. Such approaches could, for example, be (1) designing a model from the ground up for a specific combination of catchment and study purpose rather than relying on an off-the-shelf model structure (e.g. Atkinson et al., 2002; Farmer et al., 2003; Fenicia et al., 2016), (2) quantifying model structure uncertainty through the use of model inter-comparison (e.g. Perrin et al., 2001; van Esse et al., 2013; Spieler et al., 2020), (3) setting more objective limits for when efficiency scores are considered acceptable by defining benchmarks that provide a context of minimum and maximum expected model performance (e.g. Schaefli and Gupta, 2007; Seibert, 2001; Seibert et al., 2018; Knoben et al., 2020), (4) defining which model is most appropriate through evaluation metrics that go beyond the use of aggregated efficiency scores and rely on, for example, multiple metrics or data sources (e.g. Gupta et al., 2008; Kirchner, 2006; Clark et al., 2011b, a), or (5) apply model-selection or model-averaging techniques to effectively select or combine models with the appropriate strengths for a given study purpose (e.g. Neuman, 2003; Rojas et al., 2010; Schöniger et al., 2014; Höge et al., 2019)."

To prevent repetition we removed the section "Possible follow-up teaching topics" and merged this into this new section.

**References**

Knoben, W. J. M., Freer, J. E., Peel, M. C., Fowler, K. J. A., & Woods, R. A. (2020). A brief analysis of conceptual model structure uncertainty using 36 models and 559 catchments. Water Resources Research, 56, e2019WR025975. https://doi.org/10.1029/2019WR025975

---

## Referee Report (RR1)

**Knoben and Spieler (2021) HESS Review**

This work presents a set of (open source - Octave and licensed-Matlab) computational exercises that help teach hydrological model structural uncertainty, particularly model choice as an example of structural uncertainty. As the analysis and coverage of structural uncertainty are limited, this work is a useful contribution. Below are my comments and suggestions.

1. Model adequacy is closely related but different from structural uncertainty (Gupta et al., 2011). Although the authors have indicated the limitation of statistical metrics such as KGE in diagnosing model adequacy (page 4 line 3-18), they used the term adequacy in their core objective plot (Figure 1 – lessons in the three boxes). The work (and Figure 1) is based on the relative performance of two models in two catchments. As such, 'adequate model performance' is not the right phrase to use. I suggest the use of relative terms such as 'better' and/or 'a relatively high' performance.

    Similarly, it is a stretch to use strong words such as 'appropriate' and 'accurate' (on page 4 line 26 and line 31) based on comparative analysis.

2. The manuscript needs to explain why model 'm03' performs better in the two distinctly different catchments while 'm02' performs poorly in one of the catchments. Although the manuscript mentions simulating zero flows and the basis of the models' development, it is important to briefly discuss these points directly. This may support both educators and students to articulate the causes.

3. In this work, 'calibrated' parameters are used to support the comparative analysis. But, it is important to indicate/discuss the non-uniqueness of these parameters and the interplay of parameter and structural uncertainties (Clark et al., 2011; Moges et al., 2021). As separating the two uncertainties is not always straightforward, a brief discussion with references for further reading will be helpful.

Technical comments:

1. It is good not to repetitively use the term "this section describes". If necessary, it is enough to use it once (e.g., the first case on page 3 line 15 – 20). Using this term in other places (e.g., page 3 lines, 24 - 26; page 4 line 2; page 5 line 11) is just a distraction.

2. Page 2 line 21, avoid the use of the term 'For a variety of reasons'. State a few of the reasons or rewrite the sentences.

3. Page 4 lines 19 - 31 referred the catchments and models by their CAMELS and model ID. It is better to first introduce the catchment names, ID and the models' names earlier. Perhaps, on page 3 lines 14 – 15 where the objective of the paper and the experimental designs are indicated.

**References:**

Clark, M.; Kavetski, D.; Fenicia, F. Pursuing the method of multiple working hypotheses for hydrological modeling. Water Resour. Res. 2011, 47.

Gupta, H.V.; Clark, M.P.; Vrugt, J.A.; Abramowitz, G.; Ye, M. Towards a comprehensive assessment of model structural adequacy. Water Resour. Res. 2012, 48.

Moges E, Demissie Y, Larsen L, Yassin F. Review: Sources of Hydrological Model Uncertainties and Advances in Their Analysis. *Water*. 2021; 13(1):28.

---

## Author Response (AR2)

Dear authors,

Three referees have now reviewed your revised manuscript: one of them had already seen the earlier version of your manuscript, while the two others are new reviewers. While all three reviewers find the objective of your paper laudable, their recommendations are quite varied (minor revisions, major revisions, reject). I should note that two of the reviewers have extensive teaching experience, while the third one has a bit less teaching experience. The simple/simplistic way of describing model structural uncertainties is a concern noted by at least one reviewer, as well as the small number of participants involved in the testing/evaluation of the teaching module. One reviewer, in particular, suggests that more participants be involved in the testing and evaluation of the teaching module, and that learning goals be better assessed. Another reviewer is concerned about the lack of consideration of (input/output) data uncertainty and parameter uncertainty (among other elements) in the teaching module. Would there be a way for you to better articulate the role of other sources off uncertainty in the teaching module, either through additional exercises or a case study? These issues are quite critical in the context of an educational paper focused on modelling and should be addressed, which is why I am returning your manuscript for major revisions. Do not hesitate to reach out to me if you need more time to address the concerns raised by the reviewers.

With best regards,

Genevieve Ali

Dear editor, dear reviewers,

Thank you for your (continued) effort on this manuscript. Based on your comments, we have summarized the main points of improvement of our paper as follows:

1. Evaluation of the course can be improved by increasing the number of participants and better assessment of learning goals;
2. Model structure uncertainty is discussed only in isolation and the module does not discuss data uncertainty and parameter uncertainty.

In response to the reviewers' comments, we updated our evaluation procedure to closely match current practice at the University of Saskatchewan and earlier procedures as reported in the HESS Special Issue "Hydrology education in a changing world", as well as including a direct assessment of students' understanding of our intended learning goals. We ran the course again at the TU Dresden earlier this year and collected additional student responses. We updated the main manuscript and its Supporting Materials to describe this updated procedure and its outcomes. Further details are provided in the response to Reviewer 1.

Based on the reviewers' suggestions, we quantified the impact of four different sources of uncertainty on our proposed four-way model comparison that uses single calibrated parameter sets only. Our analysis covers data uncertainty, parameter uncertainty, sampling uncertainty in objective function calculation and subjectivity in the choice of objective function. The analyses indicate that, while there is some uncertainty related to all these aspects, the relative model performance that underpins the core

of our manuscript (i.e. that both models obtain similar KGE scores in one catchment, and very different scores in another) remains visible. We added a new discussion section 4.1 (pages 12-16) that describes these analyses and their outcomes, and added a suggestion to Section 2.4 "Integration in current curriculum" that teachers discuss these other sources of uncertainty in a concluding lecture after the students have performed the exercises, to ensure that students understand that differences in model performance as measured by efficiency scores can originate from multiple different causes.

We also made various minor changes to improve the manuscript's flow and clarity.

Responses to individual reviewer comments are provided on the following pages, in blue.

Kind regards,

Wouter Knoben & Diana Spieler

**1 Reviewer comments**

**1.1 Reviewer 1**

I appreciate the efforts of the authors in revising their manuscript. Many of the previous issues have been addressed, …

Thank you for your continued effort in reviewing our paper. We appreciate the time you spent on this and your efforts in outlining where we can improve.

… but I am afraid, two rather fundamental issues became apparent now:

1) Number of participants: this information was missing before. Frankly, I was surprised to read that there were only 11 participants, of which only 4 were 'normal' students, 2 were PhD students and almost half were faculty members (just as a minor comment, later the authors write about postdocs, it seems the term faculty member was used in an unusual way here – this section has been rewritten, see below). This number is clearly very low and it would be important to evaluate the course and the different design issues on a broader basis.

As indicated in our earlier response "the typical number of people in the MSc Hydrology course at TU Dresden [is] usually between 8 and 15 students per cohort". We have re-run the course at TU Dresden in early 2022 and gained another 10 evaluation responses. For what it's worth, this is the number of responses the University of Saskatchewan considers a minimum before student evaluation results can be considered stable (see below). Please note that the course was an extracurricular activity students voluntarily added to their schedule at the end of their semester. Responses show that the group of attendees feels more prepared to deal with model structure uncertainty issues after attending the course than they did before.

The diverging understanding of what a faculty member is relates to the German academic system and was rewritten to avoid confusion. We now use the term "academic or scientific staff".

For instance, based on my personal experience I would still argue that using local catchments helps the ('normal') students to see connections between model and real world and is good for their motivation.

As stated in our previous response: "[to teach] general understanding of difficulties relating to modelling, specifically selecting a few catchments and models precisely for their ability to convey this general understanding seems logical to us. Our trial application for a German audience suggests that using catchments in the United States was not detrimental to conveying these learning objectives and appetite among the audience for inclusion of more catchments was low." Our second edition of running this course for a German audience reinforces this point.

2) The authors now added an 'evaluation' of the course. Again, numbers are small (only 8), but more importantly: I don't think the value/effect of the course can be evaluated by this kind of questions. These questions evaluate more how happy the participants were with the course, but do not really evaluate what they learned.

These two points are rather fundamental. I strongly recommend that the authors test and evaluate their course with more participants and a better assessment of the learning goals before this work is published.

In response to the reviewers' comments, we updated our evaluation procedure to closely match current practice at the University of Saskatchewan and earlier procedures as reported in the HESS Special Issue "Hydrology education in a changing world", as well as including a direct assessment of students' understanding of our intended learning goals. We ran the course again at the TU Dresden earlier this year and collected additional student responses. We updated the main manuscript and its Supporting Materials to describe this updated procedure and its outcomes. Details of the analysis we used to inform this new procedure are given below.

**Summary of approach and changes to evaluation procedure**

To inform the new evaluation procedure, we collected various examples of existing evaluation procedures to better understand how learning is typically assessed. We have based this analysis on the HESS Special Issue "Hydrology education in a changing world" and on the University of Saskatchewan's internal course evaluation procedures, in the hopes of capturing both how evaluation is handled in other published manuscripts and in more practical settings.

Evaluation procedures (if present at all) in the articles in the HESS special issue are a mix of quantitative and qualitative evaluation approaches. Quantitative approaches appear exclusively based on students' self-assessment of what they learned. Such approaches are in line with the University of Saskatchewan's procedures, which also rely on self-reported assessments.

Major changes to our evaluation procedure are as follows:

- We now used a "before" and "after" survey, which lets us assess changes in student response and relate these to course effectiveness;
- We retained the 5-point answering system, because this is the same approach used at the University of Saskatchewan, and in Habib et al. (2012) and AghaKouchak et al. (2013) – the only examples in the HESS special issue on teaching that use a quantitative approach to evaluation of their proposed educational materials;
- Our new evaluation questions contain a combination of:
    - Questions related to general competence (e.g. "Through this course I gained knowledge and confidence in the general area of hydrological modelling");
    - Self-reported assessment of changes in competence (e.g. "Before/after the course, how familiar are you with model structural uncertainty");
    - Direct author-led assessment of the intended learning outcomes (e.g. "Would you agree that a model that works well in one catchment will also work well in another catchment?" – assessed before and after attending the course).

**Evaluation procedures at the University of Saskatchewan**

The University of Saskatchewan uses an internal course evaluation system that lets students self-report their experiences with the courses they attend. Evaluation outcomes are considered "stable" if n > 10. Students are asked to answer the following questions on a 5-point numeric scale:

- The course provided me with a deeper understanding of the subject matter

- I found the course intellectually stimulating
- The instructor created an environment that contributed to my learning
- Course projects, assignments, tests, and/or exams improved my understanding of the course material
- Course projects, assignments, tests, and/or exams provided opportunity for me to demonstrate an understanding of the course material
- Online-specific:
    - Online tools used to support course activities were easy for me to use. These activities could include: accessing content, submitting assignments, completing quizzes, accessing results/grades, etc.
    - The organization of online activities in the course was clear and easy to follow.
    - The online environment enriched or strengthened my learning of the course objectives/competencies.
    - The expectations for this online/remote course were made clear.
    - The instructor maintained a regular, engaged presence during online activities throughout the course.
- Overall, the quality of my learning experience in this course was: …
- Opportunity to add open comments

**Evaluation procedures in the HESS special issue**

Underlined text indicates paper titles, the contents of which are summarized as bullet point lists.

An educational model for ensemble streamflow simulation and uncertainty analysis

Reference: (Aghakouchak, Nakhjiri and Habib, 2013)

Ensemble modelling with HBV in Matlab. Contains student feedback & evaluation. Evaluation approach:

- Very brief description of student background (2 sentences)
- Anonymous survey (n = 56), with 5 options per question (1-5). Questions:
    - As a result of your work with this education toolbox in the class, what gains did you make in each of the following?
        - Hydrologic modeling in general
        - Water budget analysis
        - Rainfall-runoff processes, their mathematical formulations and the required calculations to estimate the flood resulting from a given precipitation event
        - The effect of evapotranspiration on rainfall-runoff processes, its mathematical formulation and the required calculations
        - Model calibration and ensemble simulation
        - Sensitivity analysis
        - Differences between empirical and physically-based parameters
        - Enthusiasm for the subject of hydrologic modeling and analysis
        - Confidence in performing hydrologic modeling
    - How each of the following aspects and attributes of the developed teaching tool contributed to your learning gains?
        - The use of a practical case study with actual data

- The use of hands-on calculations in the lecture
- The fact that you could change the model parameters and their effects
- The requirement of a hydrologic modeling project using this hands-on toolbox

HydroViz: design and evaluation of a Web-based tool for improving hydrology education

Reference: (Habib et al., 2012)

Online educational tool. Contains evaluation (n_student = 182, n_teacher = 6). Evaluation questions:

- How effective is the conceptual design and software features of HydroViz in facilitating students' learning and delivering the embedded educational contents on hydrologic concepts and related skills?
- What are students' perceptions of various features and characteristics of HydroViz?
- What are students' perceptions of HydroViz as a part of the curriculum?
- How effective is HydroViz in developing freshmen engineering students' interest in hydrology as a subject area?
- Do students in different classes and universities differ in their learning of the hydrologic concepts and perceptions of HydroViz?
- What can be done to improve HydroViz?

Answer to questions obtained from:

- Tasks given to students as part of exercises and correctness of answers assessed;
- Online survey where students were asked to quantify their own knowledge gains on a 5-point scale, using 17 statements;
- Informal interviews

Computer-supported games and role plays in teaching water Management

Reference: (Hoekstra, 2012)

Board game about stakeholder interaction. No evaluation to speak of.

Web 2.0 collaboration tool to support student research in hydrology – an opinion

Reference (Pathirana, Gersonius and Radhakrishnan, 2012)

Wiki for use during thesis projects. Evaluation through qualitative "Students' / educators' impressions" (n_student = 5, n_teacher = 1?), based on an interview of 5 open questions. Not anonymous. Full responses in supplement.

Water management simulation games and the construction of knowledge

Reference: (Rusca, Heun and Schwartz, 2012)

Ravilla simulation game to introduce people to Integrated Water Resource Management problems. Qualitative evaluation only.

Teaching hydrological modeling with a user-friendly catchment-runoff-model software package

Reference: (Seibert and M. J.P. Vis, 2012)

Stand-alone HBV model with suggested exercises. No formal evaluation performed. Evaluation limited to a few qualitative statements in conclusions.

Irrigania – a web-based game about sharing water resources

Reference: (Seibert and M. J. P. Vis, 2012)

Browser game about stakeholder interaction. No formal evaluation of learning goals, apart from qualitative statements about observed behavior shown by students.

**1.2    Reviewer 3**

I have been pondering about this manuscript for quite a while and I am still not sure what really to make of it. Quite clearly, the authors are right in underlining the importance of model structure uncertainty or, more pointedly, our past (and ongoing) failure to formulate general, catchment-scale theories (and thus models) from available data (cf. Nearing et al., 2021). Similarly, I agree with the authors that there are multiple (interlaced) facets to uncertainty, many of which are far from straightforward to grasp – in particular for many students at Master level.

Thank you for these positive thoughts about the usefulness of our work.

However, I am nevertheless surprised by the suggestions made by the authors and I am concerned that they could end up doing disservice to our students. The reason for this is the overly simplistic and informal – almost leisurely – way to define, explain and demonstrate uncertainties in the suggested experiment.

As I understand it, the experiment consists of a simple application of a modular modelling framework, in which 2 models are selected (why these ones?) …

As described in Section 2.1, *"[b]oth models and catchments have been specifically selected out of a sample of 40+ models and 500+ catchments for the lessons that can be conveyed by each comparative exercise."* The chosen models and catchments, when calibrated against streamflow observations from both catchments, achieve the Kling-Gupta Efficiency scores shown in Figure 1. Four-way comparison of these scores leads students to our chosen learning objectives.

… and their runs with optimum(?) parameter sets are compared.

This is correct and you are right to point out that this was not particularly clear in the earlier version of this manuscript. We have clarified the caption of Figure 1 and the corresponding text in section 2.1 to specifically mention that this concerns a single calibrated parameter set, and that calibration is performed by the students using adaptations of existing scripts in the MARRMoT repository.

How, in the absence of (1) any quantification of uncertainties related to data (or the use thereof), (2) any meaningful quantification of the uncertainties in the parameters or (3) any considerations of the impact of the choice of performance metric, can the authors suggest that the differences in the models' skill to reproduce stream flow is indeed linked to model structural errors? I believe that this conveys a way too simplistic view of uncertainty to the students. What will they take from such an example? Without any representation of the other sources of uncertainty, the risk is that many students may learn from that example that all the differences between the models are due to uncertainty in structure of the deterministic model. That is of course wrong.

Although I welcome the authors very laudable intention to sensibilize students for different sources of uncertainty, I believe that this requires a much more in-depth analysis.

It is good to know you see merit in this particular piece of work. You are right that central to our manuscript is a four-way comparison of the performance of calibrated parameter sets for two models in two catchments. In one catchment, models perform similarly in terms of KGE scores while in the other catchment model performance is very different. Based on your comments, we investigated:

- The impact of data uncertainty through modifying the data used for model calibration and evaluation;
- The impact of parameter uncertainty through:
  - First confirming through Latin Hypercube sampling that our calibration algorithm returns solutions in those regions where highest model performance is found through sampling;
  - Then assessing whether using any of the 100 best parameter sets identified through sampling would substantially alter relative model performance in either catchment.
- The impact of sampling uncertainty in calculation of the objective function;
- The impact of using a different objective function.

The analyses indicate that, while there is some uncertainty related to all these aspects, the relative model performance that underpins the core of our manuscript (i.e. that both models obtain similar KGE scores in one catchment, and very different scores in another) remains visible.

The analyses summarized above are discussed in the new section 4.1 (pages 12-16).

We added a suggestion to Section 2.4 "Integration in current curriculum" that teachers discuss these other sources of uncertainty in a concluding lecture after the students have performed the exercises, to ensure that students understand that differences in model performance as measured by efficiency scores can originate from multiple different causes.

Please also note that although I am teaching hydrology at Master level (including classes and hands-on examples on different uncertainties), I do not consider myself as educational expert. I therefore cannot provide a valid assessment of the educational value of the suggested experiment that goes beyond my hydrology-related concerns about the experiment and my 20 years in-class experience.

**1.3 Reviewer 4**

This is a useful contribution towards teaching structural uncertainty based on model comparative analysis. Please follow my detailed comments in the attachment.

This work presents a set of (open source - Octave and licensed-Matlab) computational exercises that help teach hydrological model structural uncertainty, particularly model choice as an example of structural uncertainty. As the analysis and coverage of structural uncertainty are limited, this work is a useful contribution. Below are my comments and suggestions.

*Thank you for your time and the comments on our work. It is encouraging to read that you see merit in this work.*

1. Model adequacy is closely related but different from structural uncertainty (Gupta et al., 2011). Although the authors have indicated the limitation of statistical metrics such as KGE in diagnosing model adequacy (page 4 line 3-18), they used the term adequacy in their core objective plot (Figure 1 – lessons in the three boxes). The work (and Figure 1) is based on the relative performance of two models in two catchments. As such, 'adequate model performance' is not the right phrase to use. I suggest the use of relative terms such as 'better' and/or 'a relatively high' performance.
   Similarly, it is a stretch to use strong words such as 'appropriate' and 'accurate' (on page 4 line 26 and line 31) based on comparative analysis.
   *Thank you for this comment. We agree with this sentiment and have made multiple changes throughout the manuscript (including Figure 1) to be more precise in our use of language. We decided to retain the use of the word "accurate" as this describes the mathematical (mis)match between simulations and observations. We added a new sentences to Section 2.1 to explicitly define these terms:*

   *"In the remainder of this work, we refer to the calculation of efficiency scores as the accuracy of a model's simulation, in the sense that simulations with higher efficiency scores more accurately resemble observations than the simulations from models with lower efficiency scores. This is contrasted by the term adequacy which is more commonly used to refer to a model's degree of realism (see e.g. Gupta et al., 2011)."*

2. The manuscript needs to explain why model 'm03' performs better in the two distinctly different catchments while 'm02' performs poorly in one of the catchments. Although the manuscript mentions simulating zero flows and the basis of the models' development, it is important to briefly discuss these points directly. This may support both educators and students to articulate the causes.
   *We agree that such a discussion would clarify the manuscript. We have added the necessary explanations to Section 3.2 where the relevant exercise is discussed.*

3. In this work, 'calibrated' parameters are used to support the comparative analysis. But, it is important to indicate/discuss the non-uniqueness of these parameters and the interplay of parameter and structural uncertainties (Clark et al., 2011; Moges et al., 2021). As separating the two uncertainties is not always straightforward, a brief discussion with references for further

reading will be helpful.

We investigated the impact of various sources of uncertainty on our proposed exercise and added those findings to the new Section 4.1. We also added a suggestion in Section 2.3 that teachers discuss these concepts in a concluding lecture after the students completed the proposed exercise, so that the students may be in a better position to appreciate these (reasonably complex) concepts.

Technical comments:

1.  It is good not to repetitively use the term "this section describes". If necessary, it is enough to use it once (e.g., the first case on page 3 line 15 – 20). Using this term in other places (e.g., page 3 lines, 24 - 26; page 4 line 2; page 5 line 11) is just a distraction.

    Thank you for this comment. We have found that descriptions such as these are useful to manage reader expectations – particularly at the start of section 2 - but agree that we may have overdone things a bit. We removed the mentions on page 4 line 2 and page 5 line 11 to streamline the text.

2.  Page 2 line 21, avoid the use of the term 'For a variety of reasons'. State a few of the reasons or rewrite the sentences.

    We have rewritten these sentences as follows: *"Regrettably, suitability of a given model for the task at hand is not always the main driver in model selection. Prior experience with a given model combined with lacking insights into model strengths and weaknesses often lead to a certain attachment of hydrologists to their model of choice (Addor and Melsen, 2019). Hands-on experience with model structure uncertainty in a classroom setting, particularly through exercises that show that the choice of model can have a strong impact on the quality of simulations for a given catchment, will prepare students to think beyond their 'model of choice'. This will prepare students for when they will need to design modeling studies or interpret modeling results in their future careers."*

3.  Page 4 lines 19 - 31 referred the catchments and models by their CAMELS and model ID. It is better to first introduce the catchment names, ID and the models' names earlier. Perhaps, on page 3 lines 14 – 15 where the objective of the paper and the experimental designs are indicated.

    We moved the sentences used to introduce "Section 2.2 Catchments and models" to the suggest location, and so ensure that the reader knows the catchment and model names and IDs before the reach the learning goals in Section 2.1.

References:

Clark, M.; Kavetski, D.; Fenicia, F. Pursuing the method of multiple working hypotheses for hydrological modeling. Water Resour. Res. 2011, 47.

Gupta, H.V.; Clark, M.P.; Vrugt, J.A.; Abramowitz, G.; Ye, M. Towards a comprehensive assessment of model structural adequacy. Water Resour. Res. 2012, 48.

Moges E, Demissie Y, Larsen L, Yassin F. Review: Sources of Hydrological Model Uncertainties and Advances in Their Analysis. Water. 2021; 13(1):28

**2 References**

Aghakouchak, A., Nakhjiri, N. and Habib, E. (2013) 'An educational model for ensemble streamflow simulation and uncertainty analysis', *Hydrology and Earth System Sciences*, 17(2), pp. 445–452. doi:10.5194/hess-17-445-2013.

Habib, E. *et al.* (2012) 'HydroViz: design and evaluation of a Web-based tool for improving hydrology education', *Hydrology and Earth System Sciences*, 16(10), pp. 3767–3781. doi:10.5194/hess-16-3767-2012.

Hoekstra, A.Y. (2012) 'Computer-supported games and role plays in teaching water management', *Hydrology and Earth System Sciences*, 16(8), pp. 2985–2994. doi:10.5194/hess-16-2985-2012.

Pathirana, A., Gersonius, B. and Radhakrishnan, M. (2012) 'Web 2.0 collaboration tool to support student research in hydrology – an opinion', *Hydrology and Earth System Sciences*, 16(8), pp. 2499–2509. doi:10.5194/hess-16-2499-2012.

Rusca, M., Heun, J. and Schwartz, K. (2012) 'Water management simulation games and the construction of knowledge', *Hydrology and Earth System Sciences*, 16(8), pp. 2749–2757. doi:10.5194/hess-16-2749-2012.

Seibert, J. and Vis, M. J. P. (2012) 'Irrigania – a web-based game about sharing water resources', *Hydrology and Earth System Sciences*, 16(8), pp. 2523–2530. doi:10.5194/hess-16-2523-2012.

Seibert, J. and Vis, M. J.P. (2012) 'Teaching hydrological modeling with a user-friendly catchment-runoff-model software package', *Hydrology and Earth System Sciences*, 16(9), pp. 3315–3325. doi:10.5194/hess-16-3315-2012.